# Biological insights from multi-omic analysis of 31 genomic risk loci for adult hearing difficulty

Gurmannat Kalra[1,2], Beatrice Milon[3], Alex M. Casella[1,2,4], Brian R. Herb[1],
Elizabeth Humphries[1,5], Yang Song[1], Kevin P. Rose[2,3], Ronna Hertzano[1,3,6]*, Seth
A. Ament[1,7]*

**1** Institute for Genome Sciences, University of Maryland School of Medicine, Baltimore, MD, United States of
America, **2** Program in Molecular Medicine, University of Maryland School of Medicine, Baltimore, MD, United
States of America, **3** Department of Otorhinolaryngology-Head & Neck Surgery, University of Maryland
School of Medicine, Baltimore, MD, United States of America, **4** Physician Scientist Training Program,
University of Maryland School of Medicine, Baltimore, MD, United States of America, **5** Program in Molecular
Epidemiology, University of Maryland School of Medicine, Baltimore, MD, United States of America,
**6** Department of Anatomy and Neurobiology, University of Maryland School of Medicine, Baltimore, MD,
United States of America, **7** Department of Psychiatry, University of Maryland School of Medicine, Baltimore,
MD, United States of America

* rhertzano@som.umaryland.edu (RH); sament@som.umaryland.edu (SAA)

**Data Availability Statement:** ATAC-seq (GSE126129), mRNA-seq (GSE158325), and scRNA-seq (GSE135737) data have been deposited in the Gene Expression Omnibus. Bed

## Abstract

Age-related hearing impairment (ARHI), one of the most common medical conditions, is strongly heritable, yet its genetic causes remain largely unknown. We conducted a meta-analysis of GWAS summary statistics from multiple hearing-related traits in the UK Biobank (n = up to 330,759) and identified 31 genome-wide significant risk loci for self-reported hearing difficulty (p < 5x10$^{-8}$), of which eight have not been reported previously in the peer-reviewed literature. We investigated the regulatory and cell specific expression for these loci by generating mRNA-seq, ATAC-seq, and single-cell RNA-seq from cells in the mouse cochlea. Risk-associated genes were most strongly enriched for expression in cochlear epithelial cells, as well as for genes related to sensory perception and known Mendelian deafness genes, supporting their relevance to auditory function. Regions of the human genome homologous to open chromatin in epithelial cells from the mouse were strongly enriched for heritable risk for hearing difficulty, even after adjusting for baseline effects of evolutionary conservation and cell-type non-specific regulatory regions. Epigenomic and statistical fine-mapping most strongly supported 50 putative risk genes. Of these, 39 were expressed robustly in mouse cochlea and 16 were enriched specifically in sensory hair cells. These results reveal new risk loci and risk genes for hearing difficulty and suggest an important role for altered gene regulation in the cochlear sensory epithelium.

## Author summary

The genetic architecture of age-related hearing impairment (ARHI), a strongly heritable condition, has not been well studied. We present a systems genetics analysis of risk loci for ARHI. We performed a joint GWAS analysis of four hearing related traits from the

files depicting ATAC-seq peaks, processed and annotated single-cell RNA-seq data, and GWAS summary statistics from the MTAG GWAS meta-analysis are available on the Neuroscience Multi-Omic Archive (NeMO Archive), an NIH-funded genomics data repository for the BRAIN Initiative, at: http://data.nemoarchive.org/other/grant/sament/sament/hearing_gwas/. A web browser enabling visualization and analysis of the scRNAseq, mRNA-seq, ATAC-seq, and hearing difficulty GWAS data is available on gEAR (umgear.org) within the profile: https://umgear.org/p?l=3a70e6e7. All other relevant data are within the manuscript and its Supporting Information files.

**Funding:** This project was supported by National Institute of Mental Health, BRAIN Initiative (R24 MH114815, R.H.) (https://projectreporter.nih.gov/project_info_description.cfm?aid=9766403&icde=49381344&ddparam=&ddvalue=&ddsub=&cr=3&csb=default&cs=ASC&pball=); National Institute of Deafness and Other Communication Disorders (R01 DC013817, R.H.) (https://projectreporter.nih.gov/project_info_description.cfm?aid=9654735&icde=49381326&ddparam=&ddvalue=&ddsub=&cr=1&csb=default&cs=ASC&pball=); National Institute of Deafness and Other Communication Disorders (R00 DC013107, Thomas Coate, PI) (https://projectreporter.nih.gov/project_info_description.cfm?aid=9304158&icde=49381320&ddparam=&ddvalue=&ddsub=&cr=1&csb=default&cs=ASC&pball=); Hearing Health Foundation Hearing Restoration Project, Project #5 (S.A.A.) (https://hearinghealthfoundation.org/hearing-restoration-project); Hearing Health Foundation Hearing Restoration Project, Project #4 (R.H.) (https://hearinghealthfoundation.org/hearing-restoration-project). The funders had no role in study design, data collection and analysis, decision to publish, or preparation of the manuscript.

**Competing interests:** The authors have declared that no competing interests exist.

UK Biobank and identified 31 genome-wide significant risk loci for hearing difficulty, eight of which have not been previously reported. By integrating these risk loci with transcriptomic and epigenomic data from the mouse cochlea, we discovered that risk loci are strongly enriched at genes and open chromatin regions that are active in cochlear sensory epithelial cells. Our results suggest an important role in ARHI for altered gene regulation in cochlear hair cells and supporting cells.

## Introduction

Age-related hearing impairment (ARHI) is characterized by a decline of auditory function due to impairments in the cochlear transduction of sound signals and affects approximately 25% of those aged 65–74 and 50% aged 75 and older[1]. Causes of ARHI and related forms of adult-onset hearing difficulty include a complex interplay between cochlear aging, noise exposure, genetic predisposition, and other health co-morbidities. Anatomical and physiological evidence suggest that these forms of hearing difficulty arise most commonly from damage to cochlear sensory epithelial cells, particularly inner and outer hair cells. Some forms of hearing difficulty also arise from damage to non-epithelial cells in the cochlea, including spiral ganglion neurons and cells of the stria vascularis.

Twin and family studies suggest that 25–75% of risk for ARHI is due to heritable causes[2]. Mutations in >100 genes cause monogenic deafness or hearing loss disorders[3]. However, a substantial fraction of patients with ARHI do not have a mutation in any known deafness gene, suggesting that additional genetic causes remain to be discovered. Common genetic variation may contribute to these unexplained cases of hearing difficulty, but specific risk variants remain poorly characterized. Genome-wide association studies (GWAS) of hearing-related traits, including ARHI, tinnitus, and increased hearing thresholds, have identified in aggregate approximately 50 genome-wide significant risk loci[4–8], with the largest study of adult hearing difficulty reporting 44 risk loci[9]. Positional candidate genes at these risk loci include *TRIOBP*, a gene associated with prelingual nonsyndromic hearing loss[4]; *ISG20*, encoding a protein involved in interferon signaling[4]; *PCDH20*, a member of the cadherin family[5]; and *SLC28A3*, a nucleoside transporter[5]. In addition, several studies have reported suggestive associations near *GRM7*, encoding a metabotropic glutamate receptor[6,8].

The identification of risk loci for ARHI is merely the first step toward understanding the biological mechanisms by which variants at these loci influence hearing loss. The majority of GWAS risk loci that have been identified for ARHI contain no protein-coding SNPs, making it difficult to infer the causal genes. These findings are consistent with GWAS of other common traits, which have further demonstrated that very frequently the gene closest to the risk-associated SNPs is not causal[10–13]. A logical hypothesis is that the causal risk variants at many of these loci influence gene regulation. However, this hypothesis has not been tested in the context of ARHI, largely because the effects of gene regulatory variants are often cell type-specific, and there is very little information about the locations of enhancers and promoters in relevant cell types in the human cochlea.

Here, we generated mRNA-seq, ATAC-seq, and single-cell RNA-seq from cochlear epithelial and non-epithelial cells of neonatal mice, and we used these data to predict causal variants and genes and disease-relevant cell types at risk loci for ARHI. This analysis also utilized a new multi-trait analysis of publicly available GWAS summary statistics from hearing-related traits in the UK Biobank (n up to 330,759), which supported 31 risk loci for hearing difficulty, of which eight have not been described in peer-reviewed publications. Our results indicate that

heritable risk for hearing difficulty is enriched in genes and putative enhancers expressed in sensory epithelial cells, as well as for common variants near Mendelian hearing loss genes. Statistical and epigenomic fine-mapping most strongly supported 50 putative risk genes at these loci, predicting both protein-coding and gene regulatory mechanisms for ARHI.

## Results

### Heritability of hearing-related traits in the UK Biobank

The UK Biobank is a population-based, prospective study with over 500,000 participants in Britain, aged 40–69 years when recruited in 2006–2010[14]. The study has collected genome-wide genotyping data as well as phenotypic data for thousands of traits, including multiple hearing-related traits. Recently, Wells et al.[9] reported GWAS of hearing difficulty and hearing aid use in this population. However, the UK Biobank resource also includes additional hearing-related traits, including tinnitus, and the molecular mechanisms remain poorly characterized. Here, we applied multi-trait meta-analysis and multi-omic fine-mapping to gain insight into the genetic architecture of biological mechanisms of hearing difficulty.

As a starting point for our analysis, we considered which of the hearing-related traits in the UK Biobank have significant heritability explained by the genotyped and imputed single-nucleotide polymorphisms (SNPs). We used publicly available summary statistics from GWAS in up to 337,000 UK Biobank participants performed by the Neale lab at Massachusetts General Hospital (http://www.nealelab.is/uk-biobank/). We examined heritability of 31 manually selected hearing-related traits, including 14 self-reported traits and 17 traits derived from International Statistical Classification of Diseases and Related Health Problems (ICD-10) codes (Fig 1A; S1 Table). Four traits, all self-reported answers to survey questions, had statistically significant heritability ($h^2$) explained by genotyped and imputed single nucleotide polymorphisms (SNPs), based on linkage disequilibrium (LD) score regression (LDSC[15]), after correction for multiple testing (raw p-values $< 2.1 \times 10^{-5}$; alpha = 0.05). The most significantly heritable trait was a subject's response to the question, "Do you find it difficult to follow a conversation if there is background noise (such as TV, radio, children playing)?" (N = 330,759, prevalence = 38%, SNP heritability [$h^2$] = 0.086, $h^2$ p-value = 7.4e-65; henceforth, background noise problems). The second most heritable trait was the response to the question, "Do you have any difficulty with your hearing?" (N = 323,978, prevalence = 26%, $h^2$ = 0.076, $h^2$ p-value = 1.2e-32; henceforth, hearing difficulty/problems). Third, the response to the question, "Do you use a hearing aid most of the time?" (prevalence = 5.1%, $h^2$ = 0.093, $h^2$ p-value = 3.4e-8; henceforth, Hearing aid user). Fourth, the response to the question, "Do you get or have you had noises (such as ringing or buzzing) in your head or in one or both ears that lasts for more than five minutes at a time?" (henceforth, tinnitus, where the most heritable response was "yes, now most or all of the time"; N = 109,411, prevalence = 6.6%, $h^2$ = 0.137, $h^2$ p-value = 1.3e-7). In their published analysis of the UK Biobank hearing traits, in a separate analysis using many of these same data, Wells et al.[9] previously reported GWAS for hearing aid use and for a trait that combines responses to the hearing difficulty and background noise questions and defines cases as those who responded "yes" to both questions. The GWAS of the UK Biobank tinnitus data has not been reported previously.

We downloaded the GWAS summary statistics for the four most heritable hearing-related traits and used LDSC to study the mean $\chi^2$ statistics, estimating the proportion of inflation due to polygenic heritability versus confounding. As expected, quantile-quantile plots indicate substantial deviation of $\chi^2$ statistics from a null distribution (S1 Fig). Background noise problems (Intercept = 1.031, Int. p = $1.3 \times 10^{-3}$) and hearing difficulty (Intercept = 1.018; Int. p = $3.7 \times 10^{-2}$) had significant LDSC intercept terms, suggesting some confounding, whereas the intercept

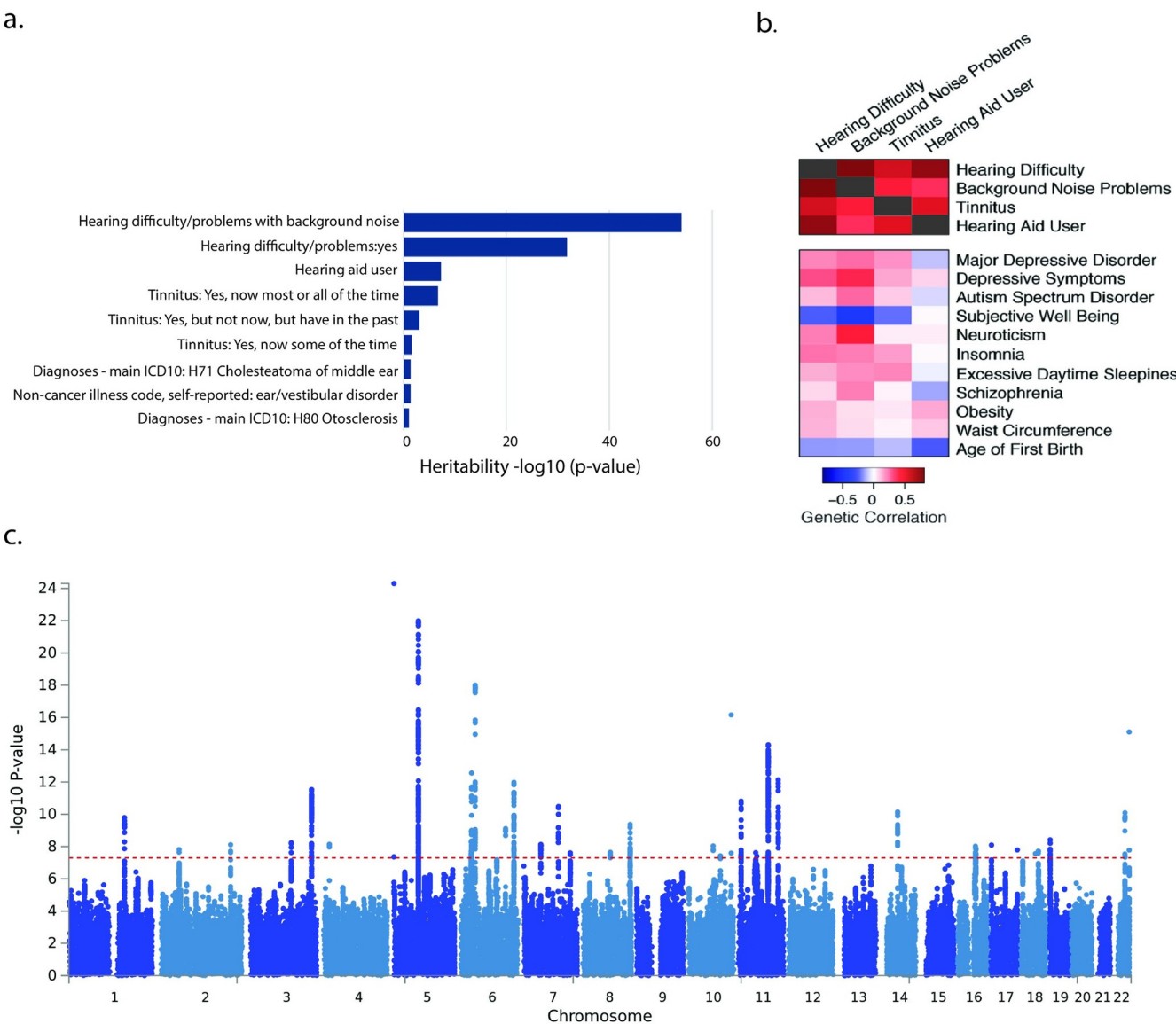

**Fig 1. Genome-wide association studies of hearing-related traits in the UK Biobank.** a. Heritability of top 9 hearing-related traits in the UK Biobank. b. Genetic correlations among the four most significantly heritable hearing-related traits and between these traits and 14 non-hearing traits that are significantly correlated with the hearing traits. c. Manhattan plot for genetic associations with hearing difficulty in the UK Biobank, following meta-analysis across the four hearing-related traits.

terms for hearing aid use and tinnitus were not significant. Reassuringly, for all four traits, LDSC intercepts ascribe >90% of the inflation in the mean $\chi^2$ to polygenic heritability rather than to confounding. These results suggest that hearing-related traits in the UK Biobank are heritable and highly polygenic. Moreover, the significant heritability and low confounding provide empirical validation for the Neale lab's analytical approach, despite certain limitations in curation and quality control that are inherent to large-scale analyses. Based on these results, we chose to use the GWAS summary statistics provided by the Neale lab for these four traits as a starting point for downstream analyses. A detailed description of these traits and analyses is provided in Methods as well as in accompanying web resources from the Neale lab and UK Biobank.

We note that the proportion of risk explained by genotyped and imputed SNPs was <10% for all of these traits, considerably less than the broad-sense heritability of ARHI based on twin and family studies. As with other complex traits, there are many potential sources for this missing heritability, including contributions from rare variants and gene x environment and gene x gene interactions. In this case, we must also consider the limitations of measuring a complex and clinically heterogeneous trait based on a small number of self-reported questions, as compared to a clinical diagnosis of ARHI based on precise measurements of auditory function.

## Genetic correlations among hearing-related and non-hearing-related traits

Next, we assessed whether risk for hearing-related traits in the UK Biobank arises from shared or distinct genetic factors. Using LDSC, we found that all pairs of hearing-related traits were genetically correlated (all $r_g \geq 0.37$; all p-values $< 7.1x10^{-8}$; S2 Table). Genetic correlations were strongest between the two most significantly heritable traits, hearing difficulty and background noise problems ($r_g = 0.81$). These results suggest substantial shared genetic architecture among these hearing-related traits.

In addition, we assessed genetic correlations between the UK Biobank hearing-related traits and GWAS of 234 non-hearing traits, available via LD Hub[16]. As expected, genetic correlations among hearing traits were stronger than genetic correlations between hearing traits and non-hearing traits. In addition, we detected significant genetic correlations (False Discovery Rate < 5%) between hearing-related traits and 14 of the 234 non-hearing traits (Fig 1B, S2 Table). Eleven of the 14 genetically correlated traits are psychiatric and personality traits, including positive genetic correlations of hearing difficulty with major depressive disorder and insomnia and a negative genetic correlation of hearing difficulty with subjective well-being. These positive genetic associations of hearing difficulty with depression-related traits is consistent with the recent report by Wells et al.[9] In addition, we detected positive genetic correlations between hearing difficulty and two anthropomorphic traits: obesity and waist circumference. Finally, we detected a negative genetic correlation between hearing difficulty and the age at first childbirth, a proxy for educational attainment and cognition. Interestingly, the genetic correlations with psychiatric and personality traits, were stronger for hearing difficulty than for hearing aid use, which could reflect differences in the underlying genetic architecture of these hearing-related traits or merely the stronger heritability of the hearing difficulty phenotype. Genetic correlations typically arise from diverse direct and indirect relationships, yet, remarkably, many of these correlations reflect known comorbidities and risk factors for hearing loss[17,18].

## Genomic risk loci and replication in independent cohorts

Leveraging the shared heritability among the four selected hearing-related traits, we performed a multi-trait analysis with MTAG[19] (Multi-Trait Analysis of GWAS). MTAG uses GWAS summary statistics from multiple traits and can boost statistical power when the traits are genetically correlated. MTAG uses bivariate LDSC to account for sample overlap between traits, making it suitable for an analysis of multiple traits measured in overlapping subjects, as in the UK Biobank. The original GWAS summary statistics for hearing difficulty included genome-wide significant associations ($p < 5x10^{-8}$) of hearing difficulty with 779 SNPs at 22 approximately LD-independent genomic loci (Fig 1C). Following joint analysis with MTAG, we identified genome-wide significant associations of hearing difficulty with 988 genotyped and imputed SNPs, located at 31 approximately LD-independent genomic loci (S2 Fig). In addition, MTAG analysis revealed 20 genome-wide significant loci for background noise problems, 25 for hearing aid use, and 20 for tinnitus (S4–S6 Tables). Most of these loci overlap the

31 loci for hearing difficulty. The p-values of the lead SNPs at most loci are strengthened in the joint analysis as compared to the individual analysis (S7 Table). In our subsequent analyses, we utilized MTAG summary statistics for hearing difficulty.

Next, we sought to replicate these findings in independent cohorts. The small sizes of available hearing-related cohorts outside UK Biobank[4–8] make them underpowered for a standard replication analysis of individual risk loci. This issue is not unique to hearing-related traits, and it has become common to report findings from biobank-scale GWAS without standard replication[20,21]. Nonetheless, these earlier cohorts provide valuable information, especially those that utilized more precise measures of hearing thresholds, which are likely to more accurately reflect hearing function than the self-reported traits in the UK Biobank. We took two approaches to replication.

First, we compared the 31 risk loci described above to the previous analysis of the UK Biobank data by Wells et al.[9]. 23 of the 31 risk loci in our analysis overlapped loci from the Wells et al. analysis. The remaining eight loci are not in LD with risk loci from Wells et al., nor with previously reported risk locus from independent cohorts. These novel risk loci are described in S3 Table, and the lead SNPs at these loci are rs2941580 (chr 2:54862003, p = 1.53 x $10^{-8}$), rs3915060 (chr 3:121712980, p = 5.84 x $10^{-9}$), rs117583072 (chr 10: 73418873, p = 1.54 x $10^{-8}$), rs61890355 (chr 11: 51422105, p = 2.39 x 10–8), rs118176061 (chr 11:54830428, p = 3.96 x 10–8), rs78417468 (chr 16:55492795, p = 1.33 x 10–8), rs118174674 (chr 18:44137400, p = 2.76 x $10^{-8}$), rs11881070 (chr 19:2389140, p = 3.85 x $10^{-9}$). Thus, meta-analysis across hearing-related traits revealed several loci that were not detected by GWAS of a more narrowly defined hearing trait.

Second, we analyzed 59 SNPs reported at genome-wide or suggestive significance levels in earlier GWAS of hearing-related traits[4–8,22] to determine whether these associations are replicated in the UK Biobank sample. Eight of these 59 SNPs showed nominally significant associations with hearing difficulty in the UK Biobank (p < 0.05; S8 Table), including both loci that reached genome-wide significance in the largest previous GWAS of ARHI[4]: rs4932196, 54 kb 3' of ISG20 (p = 2.6x$10^{-5}$ in the UK Biobank); and rs5750477, in an intron of *TRIOBP* (p = 1.3x$10^{-6}$). We note that other SNPs at the *TRIOBP* locus reached genome-wide significance in the UK Biobank. Also replicated in our analysis (notably, at genome-wide significance in the UK Biobank) were two SNPs previously reported at a suggestive significance level, in or near genes that cause Mendelian forms of hearing loss: rs9493627, a missense SNP in *EYA4*[4] (p = 7.7x$10^{-10}$); rs2877561, a synonymous variant in *ILDR1*[4] (p = 1.1x$10^{-8}$). In addition, we found a nominal level of support for rs11928865, in an intron of *GRM7*, previously reported at a suggestive significance level in multiple cohorts with ARHI[6,8] (p = 2.2x$10^{-2}$). Therefore, hearing difficulty risk variants in the UK Biobank overlap and expand upon previously discovered risk variants.

Third, we tested whether the combined, polygenic effects of hearing difficulty-associated SNPs from the UK Biobank predicted hearing ability in an independent cohort. We obtained genotypes and phenotypes from a cohort of 1,472 Belgian individuals[22] whose hearing ability was assessed with binaural thresholds for detection of low-, medium-, and high-frequency sounds. These thresholds were then summarized by principal component analysis, with PC1 giving an overall measure of hearing capabilities across all frequencies (S9 Table). A polygenic risk score[23] derived from the UK Biobank MTAG hearing difficulty summary statistics explained 1.3% of variance in PC1 from the Belgian sample (p = 7.4x$10^{-5}$; S3 Fig). The best prediction was obtained using all SNPs (GWAS p-value threshold = 1), and PRS scores using a wide range of GWAS p-value thresholds predicted a significant proportion of the variance. These results demonstrate shared genetic architecture for self-reported hearing difficulty and a more quantitative measure of hearing ability in an independent cohort.

## Heritability for hearing difficulty is enriched near Mendelian deafness genes and genes expressed in cochlear cell types

Next, we sought biological insights into hearing difficulty through gene set enrichment analyses. We performed gene-based analyses of the MTAG summary statistics using MAGMA[24] and identified 104 genes reaching a genome-wide significance threshold, $p < 2.5x10^{-6}$, correcting for 20,000 tests (S10 Table). Of these 104 genes, 40 overlap with the 31 risk loci, while the remaining genes are located at additional loci where no individual SNP reached genome-wide significance. We performed a series of hypothesis-based and exploratory gene set enrichment analyses.

It has been proposed that age-related hearing loss involves low penetrance variants in genes that are also associated with monogenic deafness disorders[4,25]. To test this hypothesis, we studied common-variant associations near 110 Mendelian deafness genes from the Online Mendelian Inheritance in Man (OMIM) database. These Mendelian deafness genes were enriched at hearing difficulty risk loci ($p = 1.19x10^{-6}$; S11 Table). We detected gene-based associations with hearing difficulty at a nominal level of significance (p-values $< 0.01$) for 15 of these 110 genes, with the strongest associations at *TRIOBP* (MAGMA: $p = 1.2x10^{-10}$), *ILDR1* ($p = 2.5x10^{-8}$), and *MYO7A* ($p = 8.5x10^{-5}$). These findings support the hypothesis that Mendelian hearing loss genes contribute to age-related hearing difficulty, but also suggest that many risk loci for hearing difficulty involve genes that have not previously been implicated in hearing loss.

A more general hypothesis is that hearing difficulty risk is enriched in genes expressed in the cochlea. We generated mRNA-seq from FACS-sorted cochlear epithelial cells, cochlear mesenchymal cells, cochlear neurons, and cochlear vascular endothelial cells from mice at postnatal day 2. We calculated the median expression of each gene in each of these cell types, as well as in subtypes of sensory epithelial hair cells and supporting cells derived from published RNA-seq[26–28]. For comparison, we considered the expression of each gene in 5,674 cell types from single-cell RNA-seq experiments of diverse mammalian tissues (S12 Table), as well as 53 extracochlear human tissues and cell types from the Genotype-Tissue Expression consortium (GTEx)[29]. Using MAGMA gene property analysis, we tested for associations of tissue-specific expression levels with genetic risk for hearing difficulty. Risk for hearing difficulty was enriched in genes expressed in cochlear epithelial cells (mostly supporting cells and hair cells; $p = 5.8e-6$), as well as in a pure population of cochlear hair cells ($p = 1.4e-5$; Fig 2; S13 and S14 Tables). Notably, enrichments of hearing difficulty risk near genes expressed in these cochlear cell types were far stronger than enrichments for brain-expressed genes, which had been reported in a previous analysis that did not include cochlear cell types[9], and for any other non-cochlear cell type (S14 Table). Therefore, our results suggest that many of the risk loci are explained by genes that are expressed in the cochlear sensory epithelium.

To identify additional functional categories enriched for hearing difficulty risk, we performed an exploratory analysis of 5,917 gene sets from Gene Ontology (GO). This analysis revealed a single significant GO term after correction for multiple testing: sensory perception of mechanical stimulus (150 genes in this set; $p = 8.62x10^{-9}$) (S15 Table). This finding is consistent with results reported by Wells et. al[9]. Taken together, these results support the relevance of hearing difficulty risk loci to the auditory system, including many genes that have not previously been associated with hearing loss.

## Heritable risk for hearing difficulty is enriched in open chromatin regions from cochlear epithelial cells

Many studies have demonstrated that GWAS associations are enriched in gene regulatory regions such as enhancers and promoters, especially in regulatory elements that are active in

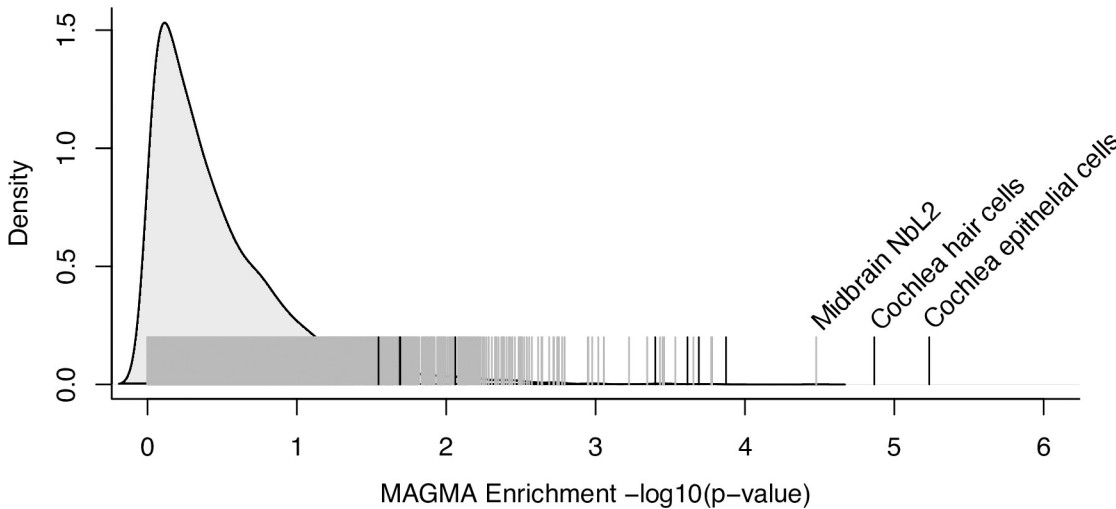

**Fig 2. Heritable risk for hearing difficulty is enriched near genes expressed in the cochlea.** Black vertical lines indicate the -log10 (p-value) for the enrichment of hearing difficulty risk near genes expressed in each cochlear cell type. Gray vertical lines indicate -log10(p-value) for genes expressed in each of 5,674 non-cochlear cell types. Labels are provided for significantly enriched cell types (p-value < 1e-4). The density plot represents the frequency distribution of p-values from non-cochlear cell types, computed using the density() function in R.

disease-relevant tissues and cell types[30,31]. Consequently, we hypothesized that SNPs influencing risk for hearing difficulty are enriched in gene regulatory regions active in the cochlea. To identify gene regulatory regions in the cochlea, we FACS-sorted epithelial cells (CD326+; including hair cells and supporting cells) and non-epithelial cells (CD326-; predominantly mesenchymal cells) from mouse cochlea at postnatal day 2 (Fig 3A), and performed ATAC-seq, on biological duplicates, to identify open chromatin regions in each cell type. Four considerations justify the use of mouse cochlea (rather than human) for this experiment: (i) mice have previously been used to successfully identify new deafness related genes and regulators of inner ear development[32–34]; (ii) at least 25% of non-coding gene regulatory elements are evolutionarily conserved between mouse and human[35]; (iii) unlike other tissues, human cochleae are not readily available for biopsies and are only rarely removed surgically; and (iv) the human cochlea (even in aborted fetuses) is already sufficiently mature to make tissue dissociation to single cells very challenging[36]. We identified 228,781 open chromatin regions in epithelial cells and 433,516 in non-epithelial cells, of which 113,733 regions were unique to epithelial cells (2.83% of the mouse genome), 320,871 unique to non-epithelial cells (4.47% of the mouse genome), and 120,919 overlapping (Fig 3B; S16 and S17 Tables). We validated these open chromatin regions through comparison to 15 experimentally validated enhancers from the VISTA Enhancer Database with activity in the ear[37] and found that ATAC-sensitive regions from both epithelial and non-epithelial cells overlapped significantly with known enhancers (epithelial cells: 3.1-fold enriched, p < $1.0x10^{-4}$; non-epithelial cells: 2.9-fold enriched, p < $1.0x10^{-4}$ based on 10,000 permutations). Examination of known cell type-specific genes suggested that chromatin accessibility in epithelial versus non-epithelial cells was correlated with cell type-specific gene expression (Fig 3C–3E). For instance, we detected open chromatin specific to epithelial cells near *Epcam* and *Sox2*, which are expressed specifically in cochlear epithelial cells[34,38]; and open chromatin specific to non-epithelial cells around *Pou3f4*, a marker for non-epithelial cells[39].

Next, we asked whether these putative regulatory regions in the cochlea are enriched for SNPs associated with hearing difficulty. Using the UCSC LiftOver tool[40], we mapped

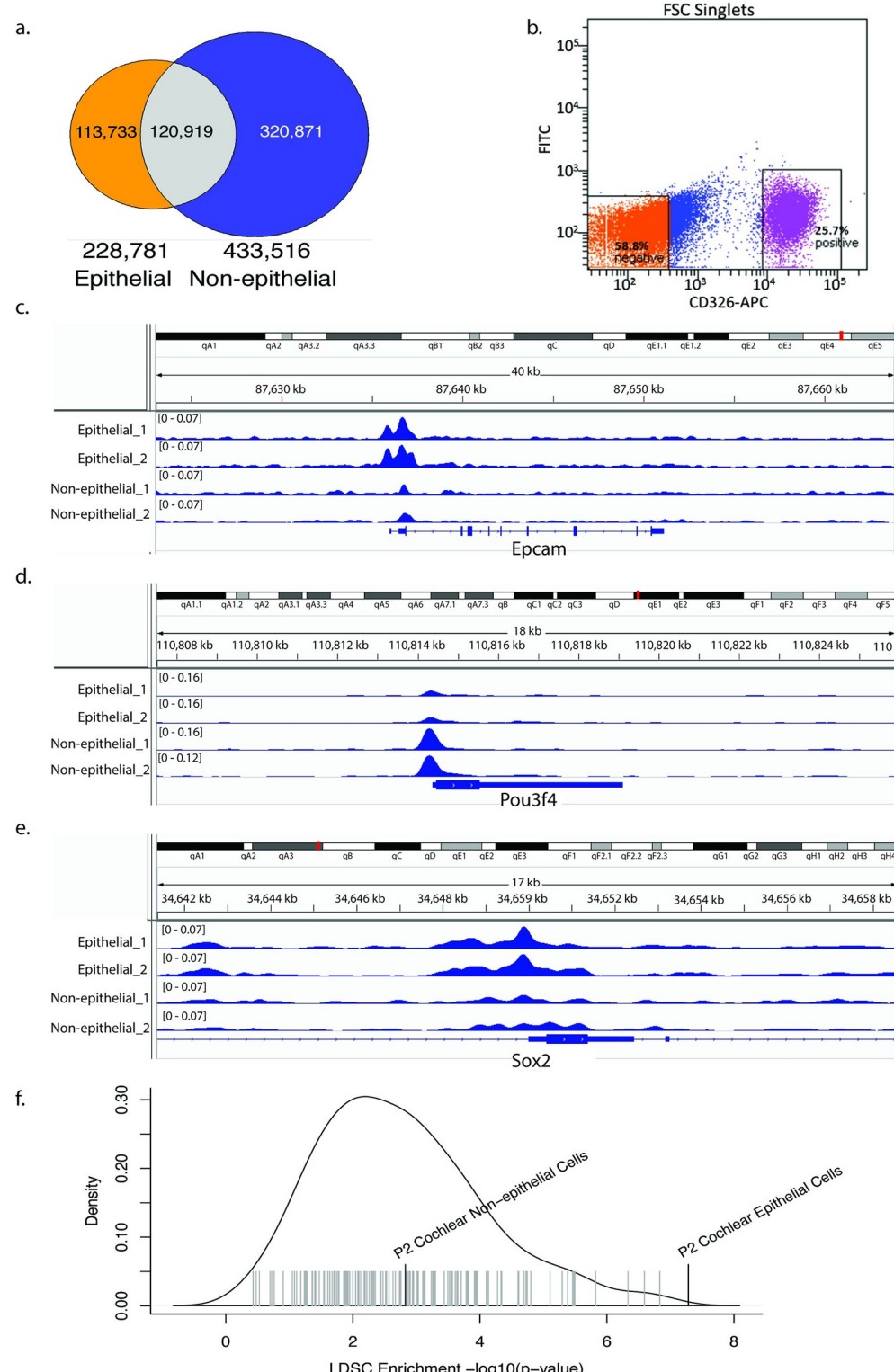

**Fig 3. Heritable risk for hearing difficulty is enriched at open chromatin regions in cochlear epithelial cells.** a. Fluorescence-activated cell sorting (FACS) of cochlear cells. Cochlear cells were labeled with a CD326 antibody conjugated to Allophycocyanin (APC), and sorted two ways as CD326 (+) and CD326 (-). b. Overlap of open chromatin regions identified by ATAC-seq of epithelial vs. non-epithelial cells in the mouse cochlea. c-e. Open chromatin peaks near cell type-specific marker genes: *Epcam* (b), *Pou3f4* (b), and *Sox2* (c). f. -log10(p-value) for

enrichment of hearing difficulty risk in regions of the human genome homologous to open chromatin in epithelial and non-epithelial cells from mouse cochlea (black vertical lines) and in non-cochlear cell types from ENCODE (gray lines) https://umgear.org/p?l=3a70e6e7.

ATAC-sensitive regions from each cochlear tissue type to the human genome to identify homologous genomic regions. 55.5% of the mouse epithelial regions and 50.2% of non-epithelial regions mapped to the human genome. To validate that these lifted-over regions are still capturing regulatory regions, we tested their overlap with ChromHMM-derived enhancers and promoters in 111 extracochlear tissues and cell types from the ROADMAP Epigenome Mapping Consortium[41]. Lifted-over regions derived from both epithelial and non-epithelial cells were substantially enriched in human promoter and enhancer regions (>10-fold and ~5-fold, respectively) but depleted in quiescent regions, heterochromatin, and gene bodies (S4 Fig, S18 and S19 Tables).

We tested for enrichment of hearing difficulty risk in these conserved regions homologous to cochlear gene regulatory regions using stratified LD score regression[42]. This model tests for heritability in cochlea-specific regions after accounting for a baseline model consisting of 24 non-cell type-specific genomic annotations, including evolutionarily conserved regions and regions that are open chromatin across many tissues. Heritable risk for hearing difficulty was enriched 9-fold in epithelial open chromatin regions (Fig 3F). 2.1% of all SNPs are in the annotated regions, and these SNPs capture 19.5% of the total SNP heritability (p = 5.2x10$^{-8}$). Heritability was less strongly–though still significantly—enriched in open chromatin regions from non-epithelial cells (4.6-fold enriched; 3.0% of all SNPs are in the annotated regions, and the SNPs capture 14.2% of the total heritability; p = 0.001). For comparison, we performed similar analyses using open chromatin regions from 147 DNase-seq experiments in 42 mouse tissues and cell types, generated by the ENCODE project[35] (S20 Table). The significance of the heritability enrichment in cochlear epithelial cells was greater than for any of the non-cochlear tissues. These results suggest that heritable risk for hearing difficulty is enriched specifically in evolutionarily conserved gene regulatory regions active in cochlear epithelial cells.

To confirm the robustness of these results, we performed comparable analyses using the original UK Biobank GWAS summary statistics for hearing difficulty as generated by the Neale lab (prior to MTAG meta-analysis); the hearing difficulty summary statistics from Wells et al.[9]; and the MTAG summary statistics for hearing aid use. In all of these analyses, we confirmed strong enrichments of heritable risk in the human genomic regions homologous to epithelial open chromatin regions, as well as more modest enrichments in the human genomic regions homologous to non-epithelial open chromatin regions (S20 Table).

## Statistical and epigenomic fine-mapping supports functional consequences to 50 genes at hearing difficulty risk loci

Next, we sought to predict causal variants and target genes at each of the 31 hearing difficulty risk loci, considering both protein-coding and putative gene regulatory consequences of each variant. To begin this analysis, we identified 613 SNPs and short indels that are in strong LD–$r^2 > 0.9$ in European samples from the 1000 Genomes project Phase 3 dataset[43]–with a genome-wide significant lead SNP at one of the 31 risk loci (S21 Table). 534 of these SNPs were included in the GWAS analysis, all of which had p-values < 2.6e-6. The remaining 79 SNPs are correlated tag SNPs for which associations were not tested directly.

We scanned these 613 risk variants for non-synonymous and stopgain SNPs, frameshift and non-frameshift indels, and effects on splice donor and acceptor sites, focusing on those variants that are predicted to be deleterious with a CADD Phred score > 10. We identified

**Table 1. Deleterious protein-coding variants in strong LD with LD-independent genome-wide significant SNPs associated with hearing difficulty.**

| Risk Locus | SNP | rsID | MAF | GWAS P-Value | Beta | r² | Ind. Sig. SNP | Gene Symbol | A.A. Change | CADD |
|---|---|---|---|---|---|---|---|---|---|---|
| 6 | 4:17524570 C/G | rs13147559 | 0.13 | 8.4e-9 | 0.02 | 0.97 | rs13148153 | CLRN2 | Leu113Val | 23.6 |
| 8 | 6:43273604 A/G | rs2242416 | 0.42 | 1.3e-18 | 0.02 | 1.00 | rs10948071 | CRIP3 | Ile188Thr | 23.8 |
| 9 | 6:133789728 A/G | rs9493627 | 0.31 | 7.7e-10 | 0.01 | 1.00 | rs9493627 | EYA4 | Gly223Ser | 26 |
| 13 | 8:82670771 A/G | rs35094336 | 0.08 | 2.5e-08 | 0.02 | 0.97 | rs74544416 | CHMP4C | Ala232Thr | 26.4 |
| 21 | 11:89017961 A/G | rs1126809 | 0.25 | 4.9e-15 | 0.02 | 1.00 | rs1126809 | TYR | Arg402Gln | 34 |
| 30 | 22:38121152 A/C | rs9610841 | 0.39 | 1.7e-10 | 0.01 | 1.00 | rs739137 | TRIOBP | Asn863Lys | 22.8 |
| 30 | 22:38122122 C/T | rs5756795 | 0.39 | NA | NA | 1.00 | rs739137 | TRIOBP | Phe1187Leu | 14.29 |
| 30 | 22:38485540 A/G | rs17856487 | 0.41 | NA | NA | 0.98 | rs132929 | BAIAP2L2 | Cys252Arg | 11.23 |
| 31 | 22:50988105 A/G | rs36062310 | 0.04 | 7.9e-16 | 0.04 | 1.00 | rs36062310 | KLHDC7B | Val504Met | 16.21 |

nine such protein-coding variants, including missense SNPs in *CLRN2*, *CRIP3*, *EYA4*, *CHMP4C*, *TYR*, *TRIOBP (2x)*, *BAIAP2L2*, and *KLHDC7B* (Table 1). Notably, six of these nine SNPs are LD-independent lead SNPs at their respective loci, increasing the statistical likelihood that these variants are causal for hearing difficulty risk. The missense SNPs in *TRIOBP* and *BAIAP2L2* are annotated to the same risk locus at 22q13.1 (S6 Fig). The two *TRIOBP* variants are in strong LD ($r^2 = 0.97$), so their effects may be additive or synergistic. By contrast, the *BAIAP2L2* variant is not in LD with either of the *TRIOBP* variants, suggesting an independent effect.

Causal variants on risk haplotypes that do not contain protein-coding variants may alter gene regulation. To elucidate these gene regulatory consequences, we annotated the 613 risk-associated SNPs in the context of the local two-dimensional and three-dimensional chromatin architecture. For the former, we utilized our ATAC-seq data from cochlear cells. For the latter, we used publicly available Hi-C data from 20 non-cochlear tissues and cell types[44]. Since data for chromatin architecture in the cochlea is unavailable, we considered all cell types in aggregate and set a stringent chromatin interaction significance threshold (p-value $< 1\text{x}10^{-25}$) to focus on the strongest chromatin loops. 126 of the 613 SNPs had potential gene regulatory functions in cochlea, based on homology to open chromatin in cochlear epithelial and non-epithelial cells from neonatal mice (S18 Table). 57 of these 126 SNPs were located proximal (<10 kb) to the transcription start sites of 17 potential target genes. In addition, 100 of the 126 SNPs could be assigned to 72 distal target genes based on long-distance chromatin loops that connect the regions containing risk-associated SNPs to these genes' transcription start sites located up to 3 Mb away (S22 Table).

We integrated the coding and non-coding functional annotations to prioritize the most likely causal genes at each locus. The union of functional annotations supported 84 genes (S23 Table). We prioritized 50 of these genes, as follows: (i) if one or more genes at a locus contained risk-associated protein-coding variants, we selected those genes; (ii) if no coding variants were identified at a locus, we selected the proximal target gene(s) of non-coding SNPs with predicted regulatory functions; (iii) if no proximal genes were identified, we considered distal target genes. This analysis identified putative risk genes at 19 of the 31 risk loci, including 10 loci at which a single gene appears most likely to be causal (S24 Table).

We sought independent support for roles of these genes in hearing loss or cochlear function based on prior evidence from genetic studies in humans and mice. Rare mutations in five of the 50 genes have been shown previously to cause Mendelian forms of deafness or hearing loss: *TRIOBP*, *EYA4*, *FTO*, *SOX2*, and *LMX1A*[4,45–53]. Genetic studies in mice have demonstrated hearing loss or cochlear development phenotypes for an additional seven of the 50 genes: *SYNJ2*, *TYR*, *PTGDR*, *MMP2*, *RPGRIP1L*, *RBL2*, and *BAIAPL2*[54–60]. Notably, three

of the five genes with independent support from human rare variants—*FTO*, *SOX2* (Fig 4A), and *LMX1A* (Fig 4B)—are located distal to risk-associated SNPs and were predicted as target genes based on long-distance chromatin interactions, validating this approach for predicting causal mechanisms. We note that there may be additional causal genes for which the functional variants are missed by our analysis. For instance, at the 3q13.3 risk locus our approach excludes a strong positional candidate, *ILDR1*, in which loss-of-function variants cause a recessive hearing loss disorder[47], since none of the risk variants at this locus were predicted to alter *ILDR1* function.

## Hearing difficulty risk genes are expressed in diverse cochlear cell types

To better understand the potential functions of the 50 putative risk genes in the cochlea, we investigated their expression patterns in cochlear cell types. We sequenced the transcriptomes of 3,411 single-cells from the mouse cochlea (postnatal day 2) using 10x Genomics Chromium technology. Cells were sequenced to a mean depth of 107,590 reads, which mapped to a median of 1,986 genes per cell. After quality control (Methods), we analyzed data from 3,314 cells. Louvain modularity clustering implemented with Seurat[61] revealed 12 major clusters of cells (Fig 5A, S5 Fig). Based on the expression of known marker genes (S25 Table), we assigned these cell clusters to the following cell types: three clusters of epithelial cells (*Epcam*+; n = 419, 101, and 24 cells per cluster), three clusters of mesenchymal cells (*Pou3f4*+; n = 887, 701, and 76 cells per cluster, of which the smallest cluster are *2810417H13Rik*+ cells undergoing cell division), 324 glial cells (*Mbp*+), 391 medial interdental cells (*Otoa*+), 59 *Oc90*+ cells, 79 vascular cells (*Cd34*+), 161 sensory epithelium supporting cells (*Sox2*+), and 91 sensory hair cells (*Pou4f3*+).

We tested cell type specific expression for each hearing difficulty risk gene. 39 of the 50 risk genes were expressed highly enough in these cochlear cells to be included in this analysis. By far the largest number of genes, 14 out of 39, were expressed selectively in sensory hair cells (Fig 5F; S26 Table). These hair cell-specific risk genes included known hearing loss genes such as *Triobp* (p = $9.5 \times 10^{-42}$) and *Eya4* (p = $2.3 \times 10^{-21}$), as well as genes that have not previously been implicated in human hearing loss; e.g., *Baiap2l2* (p = $2.6 \times 10^{-192}$; Fig 5E), *Arhgef28* (p = $9.8 \times 10^{-31}$; Fig 5C), *Gnao1* (p = $4.1 \times 10^{-29}$), *Rpgrip1l* (p = $6.3 \times 10^{-24}$), and *Crip3* (p = $9.1 \times 10^{-21}$). We also found risk genes that were expressed selectively in other cell types, including sensory epithelium supporting cells (*Sox2*, p = $9.5 \times 10^{-151}$; Fig 5D), other cochlear epithelia Oc90 cells (e.g., *Lmx1a*, p = $7.4 \times 10^{-131}$; Fig 5B), medial interdental cells (*Lpcat2*, p = $1.8 \times 10^{-139}$), and mesenchymal cells (*Mmp2*, p = $6.0 \times 10^{-120}$).

We sought to corroborate the expression pattern of risk genes in hair cells using published expression profiles from hair cells isolated by three other methods: (i) transcriptome profiling of FACS-purified hair cells versus surrounding cells[62]; (ii) translatome profiling of RiboTag-purified hair cells versus surrounding cells[63]; and (iii) single-cell RNA-seq of sensory epithelial cells[64]. Meta-analysis of these three datasets confirmed selective expression in hair cells (FDR < 0.1) for 12 of the 14 genes above (S27 Table). This analysis also revealed low but highly specific expression in hair cells for two other risk genes, *Clrn2* and *Klhdc7b*. Thus, in total, we find that 16 of the 50 putative hearing difficulty risk genes identified by GWAS are expressed selectively in sensory hair cells.

## Discussion

Here, we have described a well-powered GWAS of hearing difficulty, leveraging data from >300,000 participants in the UK Biobank and high-throughput association analyses from the Neale lab. We interpreted these genetic associations in the context of multi-omic data from the

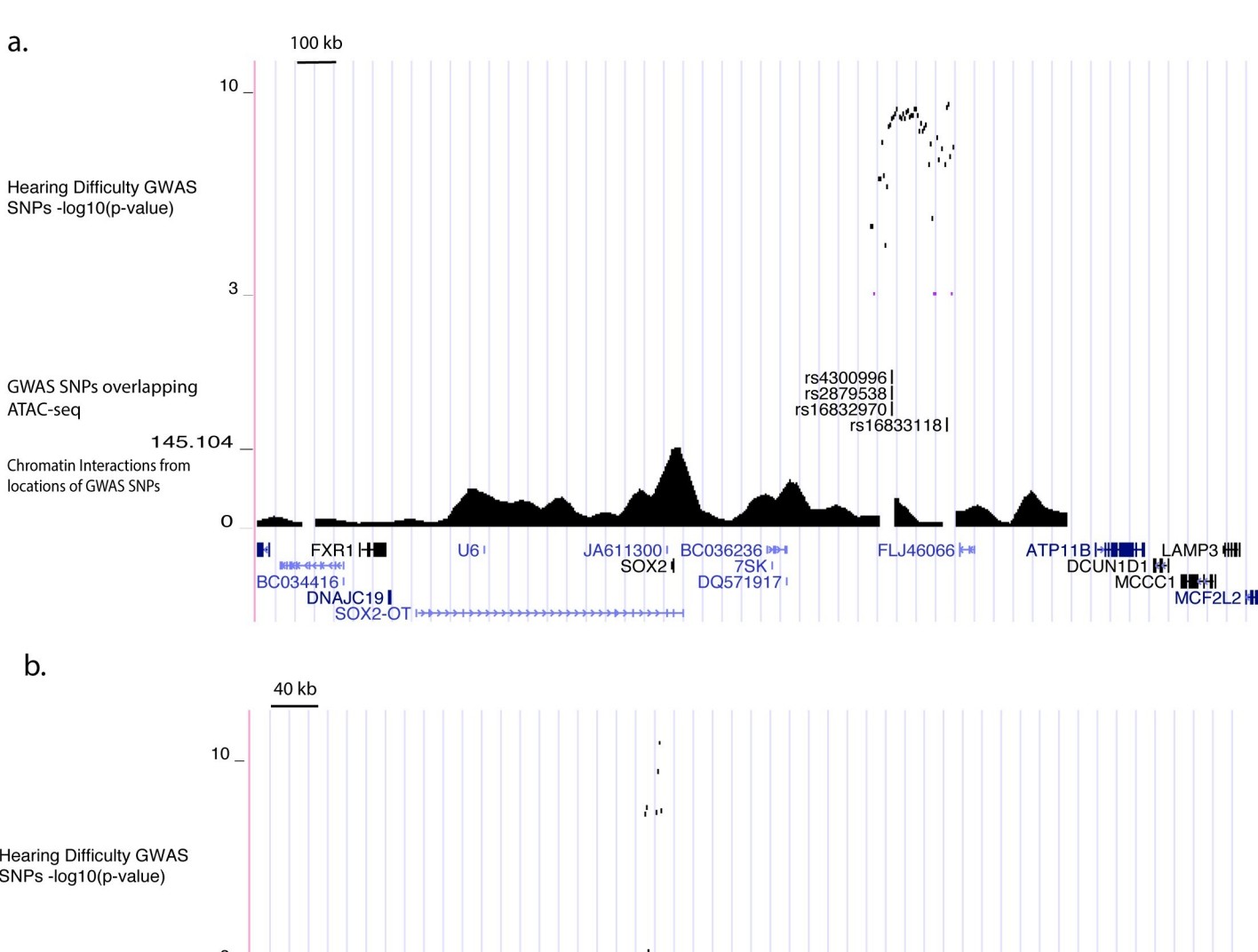

**Fig 4. Epigenomic fine-mapping predicts distal target genes for hearing difficulty risk loci.** Genetic associations and epigenomic annotations at chr3q26.3 (a) and chr1q23.3 (b). From top to bottom, genome browser tracks indicate: -log10(p-values) for association with hearing difficulty; fine-mapped SNPs in strong LD with an LD-independent lead SNP and located <500bp from a region homologous to a cochlear open chromatin region based on ATAC-seq; -log10(p-values) for chromatin interactions between the locations of the fine-mapped SNPs and distal regions, based on the minimum chromatin interaction p-value in each 40kb region from Hi-C of 20 non-cochlear human tissues and cell types[44]; locations of UCSC knownGene gene models.

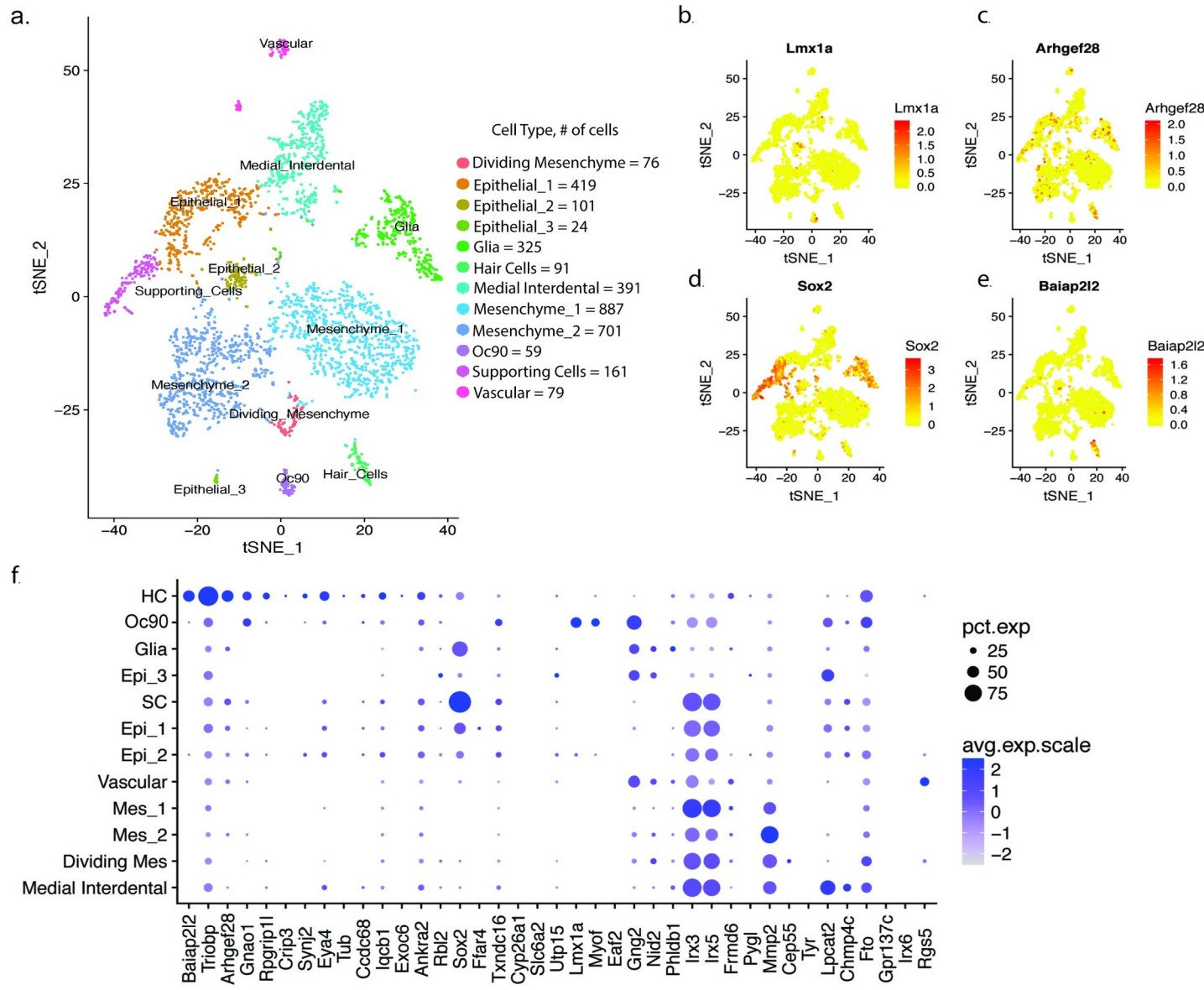

**Fig 5. Single-cell RNA-seq of mouse cochlea reveals cell type-specific expression patterns of hearing difficulty risk genes.** a. t-distributed stochastic neighbor embedding (t-SNE) plot of 3,411 cells in the postnatal day 2 mouse cochlea colored by Louvain modularity clusters corresponding to 12 cell types. b-e. t-SNE plots colored by the expression of selected hearing difficulty risk genes expressed selectively in cochlear cell types: *LMX1A* in a subset of epithelial Oc90 cells (b); *ARHGEF28* in hair cells (c), *SOX2* in supporting cells (d), and *BAIAP2L2* in hair cells (e). f. Dot plot showing the average expression and percent of cells with non-zero counts for each cochlea-expressed risk gene in each of the 12 cochlear cell types.

mouse cochlea. We identified 31 risk loci for hearing difficulty, of which eight have not been reported previously in the peer-reviewed literature. Heritable risk for hearing difficulty was enriched in genes and gene regulatory regions expressed in cochlear epithelial cells, as well as for common variants near Mendelian hearing loss genes. We identified 50 putative risk genes at these loci, many of which were expressed selectively in sensory hair cells and other disease-relevant cell types.

Genetic correlations, meta-analyses, and heritability enrichment analyses all suggest that many loci have shared effects on self-reported hearing difficulty, background noise problems,

tinnitus, and hearing aid use. This result suggests that each of the survey questions captures a similar underlying trait related to ARHI, and our MTAG meta-analysis is designed to emphasize these shared effects. However, the traits are not identical: the genetic correlations among the four traits are considerably below 1, and some distinctions were apparent in polygenic analyses. In particular, since many individuals with hearing loss do not use hearing aids, genetic factors contributing to the use of hearing aids may influence both hearing difficulty and behavioral/motivational states. In our analysis, we found a positive genetic correlation between increased risk for hearing difficulty and increased risk for major depression disorder (rg = 0.20, p = 6.0e-4), there was no significant genetic correlation between risk for hearing aid use and risk for depression (in fact, the trend is toward a negative genetic correlation; rg = -0.10, p > 0.05). ARHI is a well-known risk factor for depression in older adults. It is possible that the subset of hearing-impaired individuals who use hearing aids are genetically predisposed toward resilience or that hearing aid use ameliorates these adverse impacts. This interpretation is speculative, but the potential to detect effects of this kind motivates the continued development of large genetic cohorts for ARHI, including deeper phenotyping and questions into behavioral choices, comorbid conditions, and quality of life.

A prominent hypothesis has been that age-related hearing impairment involves lower-penetrance genetic variation in genes that cause Mendelian forms of hearing loss[4,25]. In support of this view, we found that heritability for hearing difficulty was enriched near Mendelian hearing loss genes, including genome-wide significant risk loci that overlapped three Mendelian hearing loss genes, *TRIOBP*, *EYA4*, and *ILDR1*. However, these signals represent a small fraction of the heritable risk. Indeed, our results better support a highly polygenic genetic architecture for hearing difficulty, spanning many genes that had not been known to influence hearing. Evidence for polygenicity includes the inflation of $\chi^2$ statistics (i.e., low p-values) across many thousands of SNPs and the enrichment of heritability across thousands of genes and putative gene regulatory regions expressed in the cochlear sensory epithelium. It is likely that the 31 risk loci identified here represent merely the tip of a larger genetic iceberg. Thus, as with other common traits[65], it is likely that additional risk loci and risk genes will be discovered as sample sizes for GWAS continue to grow larger.

Our results suggest that genetic risk factors for hearing difficulty act most frequently–but not exclusively—through mechanisms within sensory hair cells. The primacy of hair cells is supported by heritability enrichments for genes expressed in hair cells and for regions of open chromatin in the cochlear sensory epithelium, as well as by the fact that 16 of the 50 fine-mapped risk genes were expressed selectively in hair cells. These results strongly support the relevance of our findings to auditory function, since damage to hair cells is the most common pathophysiology in AHRI. Of the 16 hair cell-specific risk genes identified in our analysis, loss-of-function mutations in two are known to cause Mendelian hearing loss: *TRIOBP*[4,52,53] and *EYA4*[49,50]. An additional five genes have not previously been associated with human hearing loss but are known to cause hearing loss or cochlear development when mutated in mice: *SYNJ2*[54], *RPGRIP1L*[58], *BAIAP2L2*[60], *TUB*[66], and *RBL2*[59]. To our knowledge, this is the first report of a hearing loss phenotype for the remaining seven hair cell-specific risk genes: *ANKRA2*, *ARHGEF28*, *CRIP3*, *CCDC68*, *EXOC6*, *GNAO1*, *IQCB1*, and *KLHDC7B*. Hearing difficulty risk haplotypes contained protein-coding variants in 6 of these genes, while the others were supported by non-coding variants with predicted gene regulatory functions. These genes have diverse biological functions, ranging from transcriptional regulation to intracellular signaling to metabolic enzymes to structural components of synapses and stereocilia. Taken together, these results suggest that functional variants impacting a wide range of hair cell-specific genes contribute to risk for hearing loss.

Other risk genes suggest plausible mechanisms for hearing loss involving components of the stria vascularis. The lead SNP at a risk locus on chr11q14.3 is a deleterious missense SNP in *TYR*. *TYR* encodes tyrosinase, which catalyzes the production of melanin. In the cochlea, *TYR* is expressed specifically in melanin-producing intermediate cells of the stria vascularis, and *TYR* mutant mice have strial albinism, accompanied by age-associated marginal cell loss and endocochlear potential decline[55]. Chromosomal contacts at chr16q12.2 suggest that *MMP2* is targeted by two distinct risk loci. *MMP2* encodes matrix metalloproteinase-2, which serves an essential role in the cochlear response to acoustic trauma by regulating the functional integrity of the blood-labyrinth barrier[57]. Risk-associated variants in these genes may contribute to strial atrophy, a common non-sensory cause of age-related hearing impairment[67].

Perhaps more surprisingly, several of the risk genes identified in our analysis are best known for their functions in cochlear development. These include two well-characterized transcription factors: *SOX2* and *LMX1A*. Loss-of-function mutations in each of these genes cause deformations of the cochlea and hearing loss in humans[46,48], while our analyses of adult hearing difficulty revealed non-coding genetic variation in putative distal enhancers. *SOX2* is required for the formation of pro-sensory domains that give rise to hair cells and supporting cells[68]. *LMX1A* maintains proper neurogenic, sensory, and non-sensory domains in the mammalian inner ear, in part by restricting and sharpening *SOX2* expression[69]. In addition, two genes supported by putative gene regulatory interactions at the chr10q23.3 risk locus, *CYP26A1* and *CYP26C1*, metabolize retinoic acid and are involved in the specification of the otic anterior-posterior axis[70]. Changes in cochlear development may cause vulnerabilities to hearing loss later in life. Alternatively, there may be as yet undescribed roles for these genes in the adult cochlea contemporaneous with hearing loss.

Several limitations of this study should be noted. Our analysis is based on self-reported hearing difficulty, which is likely less accurate than a clinical diagnosis of ARHI or a quantitative hearing assessment. Many of the loci have not yet been replicated, which will require very large independent cohorts. The 50 genes prioritized by epigenomic fine-mapping remain data-driven hypotheses, which will need to be functionally validated in the future. Further resolving the gene regulatory interactions at these risk loci will also benefit from additional epigenomic data, including single-cell epigenomics to better resolve cell type-specificity and profiling of long-distance, 'three-dimensional" chromatin interactions through techniques such as Hi-C.

In summary, we report the first systems genetics study of common genetic variation underlying risk for adult hearing difficulty. Our multi-trait and multi-omic analyses provide novel insights into genetic architecture and molecular mechanisms, prioritizing 50 putative risk genes. Functional studies of these risk genes are warranted, especially for several hair cell-specific genes that had not previously been implicated in hearing loss. In addition, our findings support a polygenic genetic architecture for hearing difficulty, suggesting that more risk genes will be discovered as genetic data become available from additional biobank-scale cohorts.

## Methods

### Ethics statement

All procedures involving animals were carried out in accordance with the National Institutes of Health Guide for the Care and Use of Laboratory Animals and were approved by the Animal Care Committee at the University of Maryland (protocol numbers 0915006 and 1015003).

### Cell sorting by FACS followed by mRNA-seq and ATAC-seq

CD-1 timed-pregnant females were purchased from Charles River (Maryland). At postnatal day 2, the mice were euthanized and their temporal bone removed. Cochlear ducts from 20

mice were harvested, pooled and processed for Fluorescence-activated cell sorting (FACS) as described[34], for ATAC-seq. To generate cell population for mRNA-seq, single-cell suspensions were obtained from inner ears of postnatal day 0 newborn ICR mice and incubated with anti-CD326, anti-CD49f, and anti-CD34 antibodies to detect epithelial, neuronal, mesenchymal, and vascular endothelial cells. For ATAC-seq, a simplified protocol was utilized to distinguish epithelial from non-epithelial cells (primarily mesenchyme) based on labeling with anti-CD326. Cells were sorted by FACS using a BD FACS Aria II Cell Sorter (BD Biosciences) at the Flow Cytometry Facility, Center for Innovative Biomedical Resources (University of Maryland School of Medicine) (Fig 3A).

For mRNA-seq, libraries derived from total RNA from sorted cells were sequenced on an Illumina sequencer at the Institute for Genome Sciences (IGS) of the University of Maryland, School of Medicine. For ATAC-seq, fifty thousand cells and one hundred thousand cells from each sample were further processed as described [71] with the following modification: following the transposition reaction and purification step, a right side size selection (ratio 0.6) using SPRIselect (Beckman-Coulter, Indiana) was added before proceeding to the PCR amplification. This extra step resulted in the selection of DNA fragments between 150 bp to 700 bp. The following primers from [71] were used for library preparations:

```
Ad1_noMX 5'-AATGATACGGCGACCACCGAGATCTACACTCGTCGGCAGCGTCAGA
TGTG-3'; Ad2.1_TAAGGCGA 5'-CAAGCAGAAGACGGCATACGAGATTCGCCTTAG
TCTCGTGGGCTCGGAGATGT-3'; Ad2.2_CGTACTAG 5'-CAAGCAGAAGACGGCATAC
GAGATCTAGTACGGTCTCGTGGGCTCGGAGATGT-3'; Ad2.3_AGGCAGAA 5'-CAAGC
AGAAGACGGCATACGAGATTTCTGCCTGTCTCGTGGGCTCGGAGATGT-3'; Ad2.4_TCC
TGAGC 5'-CAAGCAGAAGACGGCATACGAGATGCTCAGGAGTCTCGTGGGCTCGGAGAT
GT-3'.
```
After completion of the libraries, whole genome sequencing, paired-end and a depth of 66 million reads, was performed on an Illumina HiSeq 4000 at IGS.

## Single-cell RNA sequencing of mouse cochlea

At postnatal day 2, 3 pups from a CD-1 timed-pregnant female were euthanized and their temporal bone removed. Cochlear ducts were harvested and pooled into Thermolysin (Sigma-Aldrich) for 20 min at 37°C. The Thermolysin was then replaced with Accutase (Sigma-Aldrich) and the tissue incubated for 3 min at 37°C followed by mechanical dissociation, repeating this step 3 times. After inactivation of the Accutase with 5% fetal bovine serum, the cell suspension was filter through a 35μm nylon mesh to remove cell clumps. The cell suspension was then processed for single-cell RNAseq.

Droplet-based molecular barcoding and single-cell sequencing were performed at the Institute for Genome Sciences (IGS) of the University of Maryland, School of Medicine. Approximately 10,000 dissociated cochlear cells were loaded into a Chromium Controller (10x Genomics) for droplet-based molecular barcoding of RNA from single cells. A sequencing library was produced using the 10x Single Cell Gene Expression Solution. Libraries from two cochlear samples were sequenced across three lanes of an Illumina HiSeq4000 sequencer to produce paired-end 75 bp reads.

## Description of traits and genome-wide association studies

We started by manually identifying hearing-related traits (self-reported or ICD-10 codes) in the UK Biobank Data Showcase (https://www.ukbiobank.ac.uk/data-showcase/) for which GWAS summary statistics were available in the September 20, 2017, GWAS results data release from the Neale lab (http://www.nealelab.is/blog/2017/7/19/rapid-gwas-of-thousands-of-phenotypes-for-337000-samples-in-the-uk-biobank). We used publicly available LDSC

heritability estimates for these traits produced by the Neale lab to select a smaller number for analysis. Details of the GWAS and heritability analyses are described on the Neale lab website (http://www.nealelab.is/blog/2017/9/11/details-and-considerations-of-the-uk-biobank-gwas). We identified 31 traits, shown in S1 Table. Four of these traits had significant heritability and were selected for further analysis. All of these were self-reported traits collected via ACE touchscreen questions, as follows:

1. Hearing difficulty/problems (2247; http://biobank.ctsu.ox.ac.uk/crystal/field.cgi?id=2247). Participants were asked, "Do you have any difficulty with your hearing?" Possible answers were "yes", "no", "do not know", "prefer not to answer", and "I am completely deaf". The Neale lab GWAS was performed using N = 323,978 unrelated individuals of European ancestry and compared individuals who answered yes (N = 84,839) to all other individuals (N = 239,139). We note that this analysis appears to misclassify N = 78 individuals who indicated they were completely deaf, who might better have been excluded from the analysis or treated as cases. Also, in this and other questions, it is debatable whether one should include those who answered "do not know" or "prefer not to answer." The Neale lab did not indicate how many participants included in their analysis gave these answers, but they represent 4.7% of all UK Biobank participants shown in the Data Showcase (23,333 / 498,706). Despite these potential concerns, the LDSC heritability analysis of the Neale lab GWAS indicated strong heritability (https://nealelab.github.io/UKBB_ldsc/h2_summary_2247_1.html). As the individual-level data were not available to us, we decided to use the GWAS results as provided by the Neale lab.

2. Hearing difficulty / problems with background noise (2257; http://biobank.ctsu.ox.ac.uk/crystal/field.cgi?id=2257). Participants were asked, "Do you find it difficult to follow a conversation if there is background noise (such as TV, radio, children playing)?" Possible answers were "yes", "no", "do not know", or "prefer not to answer". Participants who indicated they were previously deaf in trait 2247 were not asked this question or other subsequent hearing-related questions. The Neale lab analysis included N = 330,759 individuals and compared individuals who answered "yes" (N = 125,089) to all other individuals (N = 205,670; https://nealelab.github.io/UKBB_ldsc/h2_summary_2257.html).

3. Hearing aid user (3393; http://biobank.ctsu.ox.ac.uk/crystal/field.cgi?id=3393). Participants were asked, "Do you use a hearing aid most of the time?" Possible answers were "yes","no", or "prefer not to answer". The Neale lab analysis included $N$ = 204,240 individuals, with 10,322 answering "yes", compared to 193,918 who answered"no" or "prefer not to answer" (https://nealelab.github.io/UKBB_ldsc/h2_summary_3393.html).

4. Tinnitus, most or all of the time (4803; http://biobank.ctsu.ox.ac.uk/crystal/field.cgi?id=4803). Participants were asked, "Do you get or have you had noises (such as ringing or buzzing) in your head or in one or both ears that lasts for more than five minutes at a time?" Possible answers included "Yes, now most or all of the time", "Yes, now a lot of the time", "Yes, now some of the time", "Yes, but not now, but have in the past", "No, never", "Do not know", or "Prefer not to answer." The Neale lab performed GWAS of each of the "yes" categories, separately, in N = 109,411 individuals. The GWAS with the strongest heritability compared those indicating they experienced tinnitus most or all of the time (N = 7,214) to those who gave any other response (N = 102,197; https://nealelab.github.io/UKBB_ldsc/h2_summary_4803_11.html).

We downloaded the summary statistics from these four GWAS from the Neale lab website and used them as the starting point for subsequent analyses.

## Genetic correlations of hearing-related and non-hearing-related traits

Genetic correlations among the four hearing-related traits were calculated using the GWAS summary statistics from the Neale lab using LDSC v1.0.0. We utilized LD Scores derived from 1000 Genomes European genomes, downloaded from the LDSC website (https://data.broadinstitute.org/alkesgroup/LDSCORE/eur_w_ld_chr.tar.bz2). Genetic correlations of hearing-related traits with 234 additional traits was assessed using results from European ancestry GWAS (non-UK Biobank) of these traits available via LDHub v1.9.0 (http://ldsc.broadinstitute.org/ldhub/)[16].

## Meta-analysis of hearing-related traits in the UK Biobank

Meta-analysis of the four hearing-related traits was performed with Multi-Trait Analysis of GWAS (MTAG)[19], using a version of the MTAG software downloaded on May 22, 2018 (https://github.com/omeed-maghzian/mtag). MTAG is explicitly designed for joint analysis of summary statistics from biobank-scale GWAS of genetically correlated traits in overlapping samples. As above, this analysis used European LD Scores from LDSC and was performed using default parameters.

## Overlap of the loci identified in this study with those from the Wells et al study

The Wells et al summary statistics were downloaded from: https://zenodo.org/record/3490750#.XaXevEZKhPa. FUMA[72] v1.3.1 was used to perform functional annotation of the summary statistics. Overlap was checked by determining whether each SNP from our study was in linkage disequilibrium with any of the independent significant SNPs from the Wells et al study.

## Polygenic risk score analysis

Polygenic risk score analysis was performed to test whether hearing difficulty-associated SNPs from the UK Biobank predict hearing difficulty in an independent sample. Imputed genotypes and binaural hearing threshold phenotypes of Belgian individuals from Fransen et al.[22] were downloaded from the TGen website (https://www.tgen.org/supplementary-data/gwas_polygenic_arhi_fransen_et_al/GWAS_POLYGENIC_ARHI_Fransen_et_al.tar.gz). Initial QC was performed using PLINK v1.9. Individuals with >10% missing genotypes were filtered. We also removed SNPs that were missing from more than 10% of individuals, had a minor allele frequency of less than 5%, or were not in Hardy-Weinberg equilibrium ($p < 10^{-6}$). After filtering, 1,472 individuals and 907,726 SNPs were included in this analysis. Risk scores were calculated in these samples using the R package PRSice-2[23]. Risk scores were based on the weighted sum of risk-associated SNPs from the UK Biobank hearing difficulty MTAG summary statistics, using the following cutoffs for selection of SNPs included in the risk score: 0.001, 0.005, 0.01, 0.05, 0.1, 0.5, and 1.

## Replication of specific risk loci in independent cohorts

We tested for replication of previously reported risk loci for ARHI and other hearing-related traits through lookups in the MTAG hearing difficulty summary statistics. We started with 62 previously-reported SNPs, derived from top-level results reported by Hoffmann et al. (2016) [4], Vuckovic et al. (2015)[5], and from several earlier studies as reported in S1 Table from Ref. [4]. Summary statistics for 59 of these 62 SNPs were available in the UK Biobank sample.

## Gene set enrichment analysis

Gene set enrichment analyses were performed using MAGMA[24] v1.06 implemented within FUMA[72] v1.3.1, as well as via a standalone installation. Genotyped and imputed SNPs were annotated to ENSEMBL v92 gene models in FUMA. Annotations were limited to protein-coding genes, excluding the major histocompatibility (MHC) region of extended linkage disequilibrium (a common source of false positive results), and with SNPs mapping to a gene if they were located between the gene's start and end position. MAGMA was then used to calculate a p-value for each gene, based on the mean association among the SNPs annotated to each gene. Gene-based p-values were used to perform the following gene-level analyses:

i) Gene set enrichment analysis with Mendelian deafness genes extracted from the Online Mendelian Inheritance in Man (OMIM) database (https://omim.org/). This analysis was implemented using a local installation of MAGMA.

ii) Gene property analyses to assess covariance of MAGMA gene p-values with gene expression in six cochlear cell types, 53 non-cochlear tissues, and 5,674 cell types from single-cell RNA-seq experiments of diverse mammalian tissues. For cochlear cell types, we computed the median transcripts per million (TPM) expression level for each gene in each cell type in RNA-seq of FACS-sorted cells from GSE64543[33] and GSE60019[26]. For non-cochlear cell types, we downloaded median TPM values for 53 tissues from the GTEx v7 portal (gtexportal.org). We identified the set of genes quantified in all datasets and performed a quantile normalization across log-transformed TPM values from all cell types. Using these normalized expression levels, we performed a one-sided MAGMA gene property analysis, conditioning on the median expression level of each gene across all cell types, as well as standard covariate due to gene length, the number of SNP annotated to each gene, and correlations among nearby genes due to LD. These analyses were performed using a local installation of MAGMA. Equivalent gene property analyses of single-cell RNA-seq experiments were performed using FUMA, as described at https://fuma.ctglab.nl/tutorial#celltype.

iii) Gene set enrichment analyses for Gene Ontology terms, utilizing 6,166 gene sets from the c5.bp, c5.cc and c5.mf databases from MSigDB v5.2. This analysis was implemented in FUMA.

## ATAC-seq data processing

Four ATAC-seq fastq files (two epithelial and 2 non-epithelial samples from P1 mouse cochlea) from each tissue type were aligned to mm10 genome using BWA aligner *bwa mem* method (https://github.com/lh3/bwa). Sorted BAM files from each of the four samples were filtered to mapped reads only using samtools, converted to BED format using bedtools, and analyzed for open chromatin signal enrichment using F-Seq[73] https://umgear.org/p?l=3a70e6e7. The two BED files for each tissue type were merged using *bedtools intersect*, to identify regions common to both samples, requiring at least a 1 base pair overlap. We removed blacklist regions computed by the ENCODE project (ENCFF547MET), which show high non-specific signal across many assays.

## Determining enrichment of ATAC-seq peaks for known tissue specific enhancers

We examined overlap between open chromatin regions from ATAC-seq experiments and tissue-specific enhancers from VISTA (https://enhancer.lbl.gov/) using the Genomic Association Tester (GAT; https://github.com/AndreasHeger/gat). GAT determines the significance of overlap between genomic annotations though re-sampling within a genomic workspace defined as the mm10 genome, excluding ENCODE blacklist regions and regions of low mappability (ENCODE accession: ENCFF547MET).

## Enrichment of hearing difficulty heritability in open chromatin regions

Enrichment of hearing difficulty heritability in tissue-specific open chromatin regions from our cochlear ATAC-seq experiments, as well as from ENCODE DNase-seq experiments, was examined using stratified LDSC. 1000 Genomes Phase 3 baseline model LD scores (non tissue-specific annotations) described by Finucane, Bulik-Sullivan et al. (2015)[42] were downloaded from http://data.broadinstitute.org/alkesgroup/LDSCORE/. Open chromatin regions from DNase-seq of mouse tissues and cell types were downloaded from encodeportal.org; accession identifiers for the specific files are shown in S14 Table. Regions of the mouse genome identified in these cochlear and non-cochlear open chromatin experiments were mapped to the human genome with the UCSC Genome Browser liftOver tool, using the mm10toHg19 UCSC chain file, requiring a minimum of 50% of base pairs identical between the two genomes.

## Statistical fine mapping and functional annotation of GWAS risk loci

Fine-mapping was performed using a combination of standard annotations and analyses performed with the SNP2GENE function in FUMA v1.3.1 (http://fuma.ctglab.nl)[72], as well as additional cochlea-specific annotations downstream data integration, as described below. The architecture of risk loci was determined based on LD structure in the 1000 Genomes Phase 3 European-American sample, calculated with PLINK. LD-independent lead SNPs at each locus had p-values $< 5e-8$. We defined risk loci using a minimum pairwise $r2 > 0.6$ between lead SNPs and other SNPs. In addition, we set a minimum minor allele frequency of 0.01, and the maximum distance between LD blocks to merge into interval was 250. Subsequently, we selected 613 SNPs for deeper annotation, using a pairwise $r^2$ threshold $> 0.9$ with an LD-independent lead SNP.

Annotations of protein-coding variants were performed in FUMA, using ANNOVAR[74] with ENSEMBL v92 gene models. Deleteriousness of variants was predicted using CADD v1.3 [75], and we selected variants with a CADD Phred score threshold $> = 10$.

Regulatory functions were predicted for non-coding variants based on overlap with open chromatin in mouse cochlear epithelial and non-epithelial cells, followed by prediction of target genes based on proximity to transcription start sites (TSS) and chromatin interactions from Hi-C experiments. First, we selected a subset of the 613 risk-associated SNPs that were located +/-500bp of regions homologous to open chromatin in mouse cochlear epithelial and non-epithelial cells. SNPs were annotated to proximal target genes if they were located within 20kb of the TSS from an ENSEMBL v92 gene model.

SNPs were annotated to distal target genes based on chromatin interactions from Hi-C of 20 human tissues and cell types[44]. Hi-C data were processed with FUMA, using Fit-Hi-C[76] to compute the significance of interactions between 40kb chromosomal segments. Using these data, we identified chromosomal interactions that connect the genomic segment containing each risk-associated, open chromatin-overlapping SNP to distal chromosomal segments. We annotated genes whose transcription start sites were located within these distal segments. We considered chromosomal loops identified in each of the 20 tissues and cell types. As chromosomal contacts differ from tissue to tissue and Hi-C data are inherently noisy, aggregating loops from multiple tissues can lead to false positive signals. To mitigate this risk, we selected a strict p-value threshold for the significance of loops, $p < 1e-25$, manually determined by inspection of the data to capture one or a few of the strongest loops at each locus.

## Single cell RNA-seq data analysis

Genomic alignment, de-multiplexing, and mapping of unique molecular identifiers (UMIs) mapping to each gene was performed using cellranger. Downstream analyses were performed

with the Seurat R package[61]. We filtered cells with <50 or >20,000 UMIs and with >20% of UMIs coming from mitochondrial genes. Counts of UMIs per gene were log normalized. Highly variable genes were identified using the FindVariableGenes function with the following parameters: dispersion formula = LogVMR, minimum = 0.0123, maximum = 3, y cutoff = 0.5. Counts from variable genes were then scaled. We regressed out effects of cell cycle, percent of mitochondrial genes, and number of unique molecular identifiers. The list of cell cycle genes was obtained from the Seurat website: https://satijalab.org/seurat/cell_cycle_vignette.html. We constructed a shared-nearest neighbors graph based on the first 10 principal components of variation in the scaled and normalized expression patterns of variable genes. Cell clusters were identified from the nearest-neighbors group based on Louvain modularity, using the FindClusters() function, with a resolution of 0.6. Clusters were visualized by t-distributed sto- chastic neighbor embedding (t-SNE) of the first 10 principal components and annotated to known cochlear cell types based on the expression of all marker genes. Cell type specificity for the 50 risk genes was calculated using FindAllMarkers() function, using Wilcoxon tests to compare counts in each cell type to counts of all other cell types in aggregate.

## Meta-analysis of hair cell-specific gene expression for hearing difficulty risk genes

Processed data from GSE60019, GSE71982, and GSE116703 were downloaded and imported into R using the GEOquery R package. log-transformed transcripts per million were fit to lin- ear models, using the lmFit, contrasts.fit, and eBayes functions in the limma R package. The main effect of cell type (hair cells versus all other cells) was calculated in each dataset, sepa- rately, controlling for covariates due to age. We then computed a combined meta-analytic p- value for each gene across the three datasets, using Stouffer's z-score method with equal weights.

## Supporting information

**S1 Fig. Quantile-quantile plots of the four hearing-related traits with statistically signifi- cant heritability in the UK Biobank.** X-axis indicates the expected distribution of p-values for the associations of SNPs with each treat in the absence of true associations or confounding effects. y-axis indicates the observed distribution of p-values. A. 2247_1: "Hearing difficulty/ problems: yes" B. 2257: "Hearing difficulty/problems with background noise" C. 3393: "Hear- ing aid user" D. 4803_11: "Tinnitus: Yes, now most or all of the time."
(TIF)

**S2 Fig. Region plots for the 31 hearing difficulty risk loci (PDF document).** Each plot dis- plays the -$\log_{10}$(p-values) and genomic locations of all SNPs that are in linkage disequilibrium (LD; $r^2 > 0.6$) with an LD-independent, genome-wide significant SNP (p < 5e-8). Genes at and around each locus are shown below each plot. Eight novel loci that had not been reported in previous GWAS of hearing difficulty are indicated in bold.
(PDF)

**S3 Fig. Heritability of binaural hearing thresholds explained by hearing difficulty poly- genic risk scores in an independent sample.** Polygenic risk scores (PRS) were calculated with PRSice-2[23], defined as the weighted sum of risk-associated SNPs from the UK Biobank hear- ing difficulty MTAG summary statistics and using the p-value cutoffs indicated on the x-axis. Y-axis indicates the -$\log_{10}$(p-value) from a test of whether each PRS score predicts binaural hearing thresholds in an independent sample of 1,472 Belgian adults[22]. Binaural hearing thresholds across a range of frequencies were summarized by principal component analysis,

with principal component 1 (PC1) corresponding to the overall hearing capacity, PC2 corresponding to whether the audiogram is flat or sloping from low to high frequencies, and PC3 providing a measure of its convexity.
(TIF)

**S4 Fig. Human genomic regions homologous to open chromatin in mouse cochlea are enriched in known promoters and enhancers.** We predicted genomic regions that may be involved in gene regulation in the human cochlea based on homology to regions of open chromatin that we identified in epithelial (a) and non-epithelial cells (b) from mouse cochlea. To evaluate whether these human genomic regions correspond to true gene regulatory regions, we tested for overlap with chromatin states in 111 human tissues and cell types from the ROADMAP Epigenome Mapping Consortium. Y-axis indicates the fold enrichment (mean +/- standard error) within each chromatin state from a 25-state ChromHMM model: 1_TssA = Active TSS; 2_PromU = Promoter Upstream TSS; 3_PromD1 = Promoter Downstream TSS 1; 4_PromD2 = Promoter Downstream TSS 2; 5_Tx5 = Transcribed -5' preferential; 6_Tx = Strong transcription; 7_Tx3 = Transcribed– 3' preferential; 8_TxWk = Weak transcription; 9_TxReg = Transcribed and regulatory (Prom/Enh); 10_TxEnh5 = Transcribed 5' preferential and Enh; 11_TxEnh3 = Transcribed 3' preferential and Enh; 12_TxEnhW = Transcribed and Weak Enhancer; 13_EnhA1 = Active Enhancer 1; 14_EnhA2 = Active Enhancer 2; 15_EnhAF = Active Enhancer Flank; 16_EnhW1 = Weak Enhancer 1; 17_EnhW2 = Weak Enhancer 2; 18_EnhAc = Primary H3K27ac possible Enhancer; 19_DNase = Primary DNase; 20_ZNF_Rpts = ZNF genes & repeats; 21_Het = Heterochromatin; 22_PromP = Poised Promoter; 23_PromBiv = Bivalent Promoter; 24_ReprPc = Repressed Polycomb; 25_Quies = Quiescent/Low.
(TIF)

**S5 Fig. Expression patterns of marker genes used to identify cell types in single-cell RNA-seq of mouse cochlea.** Expression patterns of canonical marker genes used to assign cell type labels to clusters of transcriptionally similar cells in single-cell RNA-seq of postnatal day 2 mouse cochlea. X- and y-axes indicate the positions of cells in a reduced dimensional space defined by t-stochastic neighbor embedding (tSNE), with all plots here and in Fig 5 displaying the cells using the same tSNE coordinates. Canonical marker gene specificities: *Epcam*, epithelial cells; *Pou3f4*, mesenchymal cells; *Mbp*, glia (oligodendrocytes); *Pou4f3*, sensory hair cells; *Otoa*, medial interdental cells; *Oc90*+ cells; *Sox2*, sensory epithelium supporting cells; *Cd34*, vascular cells; and *2810417H13Rik*+, a marker of mesenchymal cells undergoing cell division.
(TIF)

**S6 Fig. Region plot of risk-associated SNPs at the chr22q13.1 risk locus.** Independently significant SNPs at the chr22q13.1 risk locus (rs739137 and rs132929) were in strong LD ($r^2 >$ 0.9) with three protein-coding variants in the genes *TRIOBP* and *BAIAP2L2*. rs9610841 (TRIOBP Asn863Lys) and rs5756795 (TRIOBP Phe1187Leu) were in strong LD with each other, whereas rs17856487 (BAIAP2L2 Cys252Arg) was not in LD with any SNP predicted to impact *TRIOBP*.
(TIF)

**S1 Table. Heritability of hearing-related traits.** The sample size and heritability of each hearing-related trait in the UK BioBank, based on analyses by the Neale lab at Massachusetts General Hospital. Reproduced from https://nealelab.github.io/UKBB_ldsc/.
(XLSX)

**S2 Table. Genetic correlations between hearing related traits and non-hearing related traits.** Genetic correlation between hearing-related traits and 234 non-hearing related traits measured in independent cohorts, using LDHub. For each pair of traits, we report the genetic correlation ($r_g$) and its associated p-value.
(PDF)

**S3 Table. Genomic risk loci for hearing difficulty.** Loci were defined by the set of all SNPs in LD (r2 > 0.6, 1000 Genomes Phase 3 European samples) with an independently significant SNP (p < 5e-8) at that locus. Start and end refer to the left- and rightmost positions of these SNPs (hg19 coordinates). nSNPs refers to the total number of SNPs in the locus, regardless of whether these SNPs were directly tested for association and included in the GWAS summary statistics. nGWASSNPs indicates the number of SNPs in the locus that were included in the GWAS summary statistics, a subset of "nSNPs". SNPs that are not in linkage disequilibrium with any of those identified in the previously published GWAS of hearing difficulty in the UK Biobank[9] are indicated in bold text.
(XLSX)

**S4 Table. Genomic risk loci for background noise problems.** Loci were defined by the set of all SNPs in LD (r2 > 0.6, 1000 Genomes Phase 3 European samples) with an independently significant SNP (p < 5e-8) at that locus. Start and end refer to the left- and rightmost positions of these SNPs (hg19 coordinates). nSNPs refers to the total number of SNPs in the locus, regardless of whether these SNPs were directly tested for association and included in the GWAS summary statistics. nGWASSNPs indicates the number of SNPs in the locus that were included in the GWAS summary statistics, a subset of "nSNPs".
(XLSX)

**S5 Table. Genomic risk loci for hearing aid use.** Loci were defined by the set of all SNPs in LD (r2 > 0.6, 1000 Genomes Phase 3 European samples) with an independently significant SNP (p < 5e-8) at that locus. Start and end refer to the left- and rightmost positions of these SNPs (hg19 coordinates). nSNPs refers to the total number of SNPs in the locus, regardless of whether these SNPs were directly tested for association and included in the GWAS summary statistics. nGWASSNPs indicates the number of SNPs in the locus that were included in the GWAS summary statistics, a subset of "nSNPs".
(XLSX)

**S6 Table. Genomic risk loci for tinnitus.** Loci were defined by the set of all SNPs in LD (r2 > 0.6, 1000 Genomes Phase 3 European samples) with an independently significant SNP (p < 5e-8) at that locus. Start and end refer to the left- and rightmost positions of these SNPs (hg19 coordinates). nSNPs refers to the total number of SNPs in the locus, regardless of whether these SNPs were directly tested for association and included in the GWAS summary statistics. nGWASSNPs indicates the number of SNPs in the locus that were included in the GWAS summary statistics, a subset of "nSNPs".
(XLSX)

**S7 Table. Associations of independently significant SNPs at the 31 hearing difficulty risk loci with other hearing-related traits in the UK Biobank.** Table indicates the p-values and effect sizes for each SNP in each of the four traits examined, showing results from the original GWAS performed by the Neale lab as well as from MTAG meta-analysis.
(XLSX)

**S8 Table. Replication in the UK Biobank for risk-associated SNPs from previous GWAS of hearing difficulty.** Analysis of 59 SNPs reported at genome-wide or suggestive significance

levels in earlier GWAS of hearing-related traits[4] to determine whether these associations are replicated in the MTAG analysis of hearing difficulty in the UK Biobank.
(XLSX)

**S9 Table. Polygenic risk prediction in a cohort of 1,472 Belgian individuals.** The threshold, significance, variance explained ($R^2$), effect size estimate, and the number of SNPs included in each threshold are reported.
(XLSX)

**S10 Table. MAGMA gene-based p-values.** The gene symbols, along with the chromosomal location, number of SNPs (NSNPS), number of relevant parameters used in the model (NPARAM), sample size, z-score, and p-values (raw and adjusted for multiple testing) are reported for each of the genes reaching a genome-wide significance threshold from a gene-based analyses of the MTAG summary statistics using MAGMA.
(CSV)

**S11 Table. MAGMA gene set enrichments for Mendelian deafness genes and genes expressed in cochlear cell types.** The gene Ensembl IDs, along with the chromosomal location, number of SNPs (NSNPS), number of relevant parameters used in the model (NPARAM), sample size, z-score, and p-values) are reported for each of the gene sets used in the MAGMA gene set enrichment analysis.
(XLSX)

**S12 Table. MAGMA gene set enrichments for 5,674 cell types from single-cell RNA-seq experiments of diverse mammalian tissues.** The dataset name/source, cell type that it highlights, number of genes used (NGENES), beta, standard error, and significance is reported for a MAGMA gene set enrichment of each cell type.
(XLSX)

**S13 Table. MAGMA cell specific expression.** The median transcripts per million (TPM) expression level for each gene in each cell type in RNA-seq data of FACS-sorted cells from GSE64543[33] and GSE60019[26].
(XLSB)

**S14 Table. MAGMA gene set enrichments for genes expressed in tissues from GTEx.** For each tissue, the source, the regression coefficient of the gene set, standard error, and p-value are reported.
(XLSX)

**S15 Table. GO Term enrichment of hearing difficulty risk loci.** For each GO Term, the number of genes, the regression coefficient of the gene set, standard error, the p-value, and the adjusted p-value are reported.
(XLSX)

**S16 Table. Open chromatin regions in cochlear epithelial cells.** Chromosomal locations of the open chromatin regions in cochlear epithelial cells.
(XLSX)

**S17 Table. Open chromatin regions in cochlear non-epithelial cells.** Chromosomal locations of the open chromatin regions in cochlear non-epithelial cells.
(XLSX)

**S18 Table. Overlap of conserved cochlear epithelial open chromatin regions and other cell types from ROADMAP.** We predicted genomic regions that may be involved in gene

regulation in the human cochlea based on homology to regions of open chromatin that we identified in epithelial cells from mouse cochlea. To evaluate whether these human genomic regions correspond to true gene regulatory regions, we tested for overlap with chromatin states in 111 human tissues and cell types from the ROADMAP Epigenome Mapping Consortium. Enrichment is indicated within each chromatin state from a 25-state ChromHMM model, as column headings: 1_TssA = Active TSS; 2_PromU = Promoter Upstream TSS; 3_PromD1 = Promoter Downstream TSS 1; 4_PromD2 = Promoter Downstream TSS 2; 5_Tx5 = Transcribed -5' preferential; 6_Tx = Strong transcription; 7_Tx3 = Transcribed– 3' preferential; 8_TxWk = Weak transcription; 9_TxReg = Transcribed and regulatory (Prom/Enh); 10_TxEnh5 = Transcribed 5' preferential and Enh; 11_TxEnh3 = Transcribed 3' preferential and Enh; 12_TxEnhW = Transcribed and Weak Enhancer; 13_EnhA1 = Active Enhancer 1; 14_EnhA2 = Active Enhancer 2; 15_EnhAF = Active Enhancer Flank; 16_EnhW1 = Weak Enhancer 1; 17_EnhW2 = Weak Enhancer 2; 18_EnhAc = Primary H3K27ac possible Enhancer; 19_DNase = Primary DNase; 20_ZNF_Rpts = ZNF genes & repeats; 21_Het = Heterochromatin; 22_PromP = Poised Promoter; 23_PromBiv = Bivalent Promoter; 24_ReprPc = Repressed Polycomb; 25_Quies = Quiescent/Low.
(CSV)

**S19 Table. Overlap of conserved cochlear non- epithelial open chromatin regions and other cell types from ROADMAP.** We predicted genomic regions that may be involved in gene regulation in the human cochlea based on homology to regions of open chromatin that we identified in non-epithelial cells from mouse cochlea. To evaluate whether these human genomic regions correspond to true gene regulatory regions, we tested for overlap with chromatin states in 111 human tissues and cell types from the ROADMAP Epigenome Mapping Consortium. Enrichment is indicated within each chromatin state from a 25-state ChromHMM model, as column headings: 1_TssA = Active TSS; 2_PromU = Promoter Upstream TSS; 3_PromD1 = Promoter Downstream TSS 1; 4_PromD2 = Promoter Downstream TSS 2; 5_Tx5 = Transcribed -5' preferential; 6_Tx = Strong transcription; 7_Tx3 = Transcribed– 3' preferential; 8_TxWk = Weak transcription; 9_TxReg = Transcribed and regulatory (Prom/Enh); 10_TxEnh5 = Transcribed 5' preferential and Enh; 11_TxEnh3 = Transcribed 3' preferential and Enh; 12_TxEnhW = Transcribed and Weak Enhancer; 13_EnhA1 = Active Enhancer 1; 14_EnhA2 = Active Enhancer 2; 15_EnhAF = Active Enhancer Flank; 16_EnhW1 = Weak Enhancer 1; 17_EnhW2 = Weak Enhancer 2; 18_EnhAc = Primary H3K27ac possible Enhancer; 19_DNase = Primary DNase; 20_ZNF_Rpts = ZNF genes & repeats; 21_Het = Heterochromatin; 22_PromP = Poised Promoter; 23_PromBiv = Bivalent Promoter; 24_ReprPc = Repressed Polycomb; 25_Quies = Quiescent/Low.
(CSV)

**S20 Table. Enrichments of hearing difficulty risk in open chromatin regions from cochlear and non-cochlear cell types.** Sheet 1: Enrichments of hearing difficulty risk in open chromatin regions from cochlear and non-cochlear cell types (MTAG summary statistics). Sheet 2: Enrichments of hearing difficulty risk in open chromatin regions from cochlear and non-cochlear cell types (Neale lab v1 summary statistics). Sheet 3: Enrichments of hearing difficulty risk in open chromatin regions from cochlear and non-cochlear cell types using summary statistics from the previously published GWAS of hearing difficulty in the UK Biobank[9]. Sheet 4: Enrichments of hearing aid use risk in open chromatin regions from cochlear and non-cochlear cell types (MTAG summary statistics).
(XLSX)

**S21 Table. Functional annotations of 613 SNPs in LD with LD-independent genome-wide significant SNPs at hearing difficulty risk loci.** Table indicates coding and non-coding functional annotations for 613 SNPs in LD with an LD-independent genome-wide significant SNPs at the 31 risk loci. Annotations also include predicted target genes and the evidence supporting these annotations.
(XLSX)

**S22 Table. Chromatin Interactions with hearing difficulty SNPs.** Table describes chromatin loops used to predict interactions between risk-associated SNPs and potential target genes.
(XLSX)

**S23 Table. List of likely causal genes from integrating coding and non-coding functional annotations.** For each SNP, the genomic locus, variant identifiers (rsID), effect and non-effect alleles, allele frequency, GWAS p-value, regression coefficient (beta), standard error of the beta coefficient, predicted deleteriousness based on Combined Annotation Dependent Depletion PHRED score[77] (higher is more deleterious), and gene names.
(XLSX)

**S24 Table. Annotation of 50 fine-mapped risk genes at hearing difficulty risk loci.** For each gene, the gene symbol, variant type (coding vs. non-coding), and a description.
(XLSX)

**S25 Table. Marker genes for cochlear cell types in P2 scRNA-seq data.** For each marker gene, the cell type cluster that it belongs to, as well as the p-value and the percentage of cells in that cluster that express it (pct.1) vs the percentage in all other cells (pct.2) is listed.
(XLSX)

**S26 Table. Cell type-specificity analysis for hearing difficulty risk genes.** For each gene, the top 3 clusters that it is expressed in from the single cell RNA-seq data are listed.
(XLSX)

**S27 Table. Meta-analysis of risk gene expression in hair cells vs. other cochlear cell types.** For each risk gene, the p-value and log fold change derived from each data set is reported, along with the meta-analysis p-value.
(XLSX)

## Acknowledgments

We are grateful to Benjamin Neale and the leaders of the UK Biobank for making the GWAS resource freely available to the research community, as well as to all the participants in the UK Biobank, without whom none of this work would be possible.

## Author Contributions

**Conceptualization:** Gurmannat Kalra, Ronna Hertzano, Seth A. Ament.

**Data curation:** Gurmannat Kalra, Alex M. Casella, Ronna Hertzano, Seth A. Ament.

**Formal analysis:** Gurmannat Kalra, Alex M. Casella, Brian R. Herb, Elizabeth Humphries, Yang Song, Kevin P. Rose, Seth A. Ament.

**Funding acquisition:** Ronna Hertzano, Seth A. Ament.

**Investigation:** Gurmannat Kalra, Beatrice Milon, Alex M. Casella, Kevin P. Rose, Ronna Hertzano, Seth A. Ament.

**Methodology:** Gurmannat Kalra, Beatrice Milon, Seth A. Ament.

**Project administration:** Ronna Hertzano, Seth A. Ament.

**Resources:** Gurmannat Kalra, Ronna Hertzano, Seth A. Ament.

**Software:** Gurmannat Kalra, Alex M. Casella, Brian R. Herb, Seth A. Ament.

**Supervision:** Brian R. Herb, Ronna Hertzano, Seth A. Ament.

**Validation:** Seth A. Ament.

**Visualization:** Gurmannat Kalra, Seth A. Ament.

**Writing – original draft:** Gurmannat Kalra, Seth A. Ament.

**Writing – review & editing:** Gurmannat Kalra, Alex M. Casella, Ronna Hertzano, Seth A. Ament.

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
