## [Decision Letter · Decision Letter 0]

1 Jun 2020

Dear Dr Ament,

Thank you very much for submitting your Research Article entitled 'Biological insights from multi-omic analysis of 31 genomic risk loci for adult hearing difficulty' to PLOS Genetics. Your manuscript was fully evaluated at the editorial level and by independent peer reviewers. The reviewers appreciated the attention to an important topic but identified some aspects of the manuscript that should be improved.  In addition to the comments provided by the reviewers, we ask that you include some discussion regarding the validity of "hearing aid usage".  It is well known that most (estimates are as high as 80%) people who could benefit from hearing aids will not wear them, even if they have them.  This seems to be more of an attitude issue, which could still be under genetic, albeit different genes, control.

We therefore ask you to modify the manuscript according to the review recommendations before we can consider your manuscript for acceptance. Your revisions should address the specific points made by each reviewer.

[LINK]

Yours sincerely,

Susan H. Blanton, PhD

Guest Editor

PLOS Genetics

Scott Williams

Section Editor: Natural Variation

PLOS Genetics

Reviewer's Responses to Questions

**Comments to the Authors:**

Reviewer #1: The authors have put together a comprehensive manuscript describing their GWAS using the UKBB questionnaire-based information on hearing. To my knowledge there exists no prospective data set of individuals appropriately screened and tested for ARHI that would provide the power necessary for significance. These authors, utilizing this database, have displayed a powerful tool for GWAS and have identified and replicated several genomic regions associated with ARHI. Furthermore, these authors demonstrate the homology to our best genetic model thus far, the mouse. I think this is a meaningful paper that will serve hearing research well.

Reviewer #2: Kalra et al. use bioinformatics approaches in an attempt to identify genetic loci that are associated with hearing deficits in humans. They conduct a meta-analysis of GWAS summary statistics from the UK Biobank and report the identification of previously known and 8 new putative risk loci in humans. Using mouse tissue they then carry out a variety of omics studies to identify cells in the inner ear expressing the genes, open chromatin regions, and epigenetic marks associated with the disease. The authors conclude that hearing loss in the adult is genetically complex and likely involves altered gene regulation in the cochlear sensory epithelium.

The work is generally carried out at a high standard but I am a bit conflicted how far the study advances the field conceptually. GWAS studies (at least one using the same data set) have already identified many loci associated with hearing difficulties in the adult and are replicated here. The current analysis adds genes to the list without substantially changing the overall picture. The contribution of each locus to phenotype is small, which is in agreement with earlier findings. Perhaps one of the most interesting parts of the paper is the attempt to link omics studies in mice with mapping data in humans to begin a functional characterization of the putative risk loci. Based on these data, the authors conclude that many of the genes linked to hearing loss are apparently expressed in the inner ear and in sensory structures. This is also in agreement with previous studies indicating that genetic forms of hearing loss frequently affect genes expressed in sensory structures of the inner ear. While the datasets are certainly of value for the community and worthwhile reporting, functional validation is largely missing. This is further confounded by limitations of the study. Hearing phenotypes were self-reported without quantitative assessment. Most of the identified loci have not been replicated in independent cohorts. The authors also point out that the data are a ‘best guess’ and need additional validation. Thus, while the data are certainly interesting, I find they fall short of the kind of advance expected for PLOS Genetics.

**Have all data underlying the figures and results presented in the manuscript been provided?**

Reviewer #1: Yes

Reviewer #2: Yes

PLOS authors have the option to publish the peer review history of their article (what does this mean?). If published, this will include your full peer review and any attached files.

Reviewer #1: Yes: Rick A Friedman, MD, PhD

Reviewer #2: No

---

## [Editor Report · Decision Letter 1]

4 Aug 2020

Dear Dr Ament,

We are pleased to inform you that your manuscript entitled "Biological insights from multi-omic analysis of 31 genomic risk loci for adult hearing difficulty" has been editorially accepted for publication in PLOS Genetics. Congratulations!

Yours sincerely,

Susan H. Blanton, PhD

Guest Editor

PLOS Genetics

Scott Williams

Section Editor: Natural Variation

PLOS Genetics

Comments from the reviewers (if applicable):

**Data Deposition**

http://datadryad.org/submit?journalID=pgenetics&manu=PGENETICS-D-20-00474R1

**Press Queries**

---

## [Editor Report · Acceptance letter]

23 Sep 2020

PGENETICS-D-20-00474R1 

Biological insights from multi-omic analysis of 31 genomic risk loci for adult hearing difficulty 

Dear Dr Ament, 

We are pleased to inform you that your manuscript entitled "Biological insights from multi-omic analysis of 31 genomic risk loci for adult hearing difficulty" has been formally accepted for publication in PLOS Genetics! Your manuscript is now with our production department and you will be notified of the publication date in due course.

With kind regards,

Jason Norris

PLOS Genetics

On behalf of:
